# Supercharging Imbalanced Data Learning With Energy-based Contrastive Representation Transfer

**Junya Chen**[1][*] **Zidi Xiu**[1][*], **Benjamin Goldstein**[1], **Ricardo Henao**[1],
**Lawrence Carin**[2], and **Chenyang Tao**[1]

[1]Duke University  [2] KAUST
{junya.chen, zidi.xiu, chenyang.tao}@duke.edu

## Abstract

Dealing with severe class imbalance poses a major challenge for many real-world applications, especially when the accurate classification and generalization of minority classes are of primary interest. In computer vision and NLP, learning from datasets with long-tail behavior is a recurring theme, especially for naturally occurring labels. Existing solutions mostly appeal to sampling or weighting adjustments to alleviate the extreme imbalance, or impose inductive bias to prioritize generalizable associations. Here we take a novel perspective to promote sample efficiency and model generalization based on the invariance principles of causality. Our contribution posits a meta-distributional scenario, where the causal generating mechanism for label-conditional features is invariant across different labels. Such causal assumption enables efficient knowledge transfer from the dominant classes to their under-represented counterparts, even if their feature distributions show apparent disparities. This allows us to leverage a causal data augmentation procedure to enlarge the representation of minority classes. Our development is orthogonal to the existing imbalanced data learning techniques thus can be seamlessly integrated. The proposed approach is validated on an extensive set of synthetic and real-world tasks against state-of-the-art solutions.

## 1 Introduction

Learning with imbalanced datasets is a common yet still very challenging scenario in many machine learning applications. Typical scenarios include: (*i*) rare events, where the event prevalence is extremely low while their implications are of high cost, *e.g.*, severe risks that people seek to avert [56]; (*ii*) emerging objects in a dynamic environment, which call for quick adaptation of an agent to identify new cases with only a handful target examples and plentiful past experience [31]. A typical scenario in natural datasets is that the occurrence of different objects follows a power law distribution. And in many situations, the accurate identification of those rarer instances bears more significant social-economic values, *e.g.*, fraud detection [24], driving safety [27], nature conservation [50], social fairness [17], and public health [63, 92].

Notably, severe class imbalance and lack of minority labels are the two major difficulties in this setting, which render standard learning strategies unsuitable [57]. Without explicit statistical adjustments, the imbalance induces bias towards the majority classes. On the other hand, the lack of minority representations prevents the identification of stable correlations that generalize in predictive settings. In addition, due to technological advancements, more and higher dimensional data are routinely collected for analysis. This also inadvertently exacerbates the issue of minority modeling as there is an excess of predictors relative to the limited occurrence of minority samples.

---

[*]Equal Contribution

35th Conference on Neural Information Processing Systems (NeurIPS 2021).

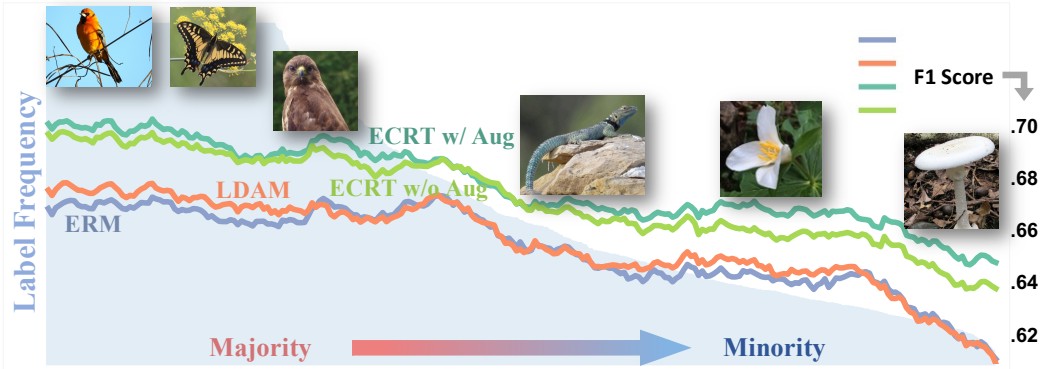

Figure 1: Energy-based causal representation transfer alleviates tail imbalance for natural image classification. The shaded region denotes label frequency in the `iNaturalist` data, with some representative images shown. Solid curves are for label-conditional F1 scores (higher is better) for the proposed *energy-based causal representation transfer* (ECRT), LDAM (state of the art) and the standard ERM. ECRT consistently outperforms the others, especially in the low-sample regime.

Various research efforts have been directed to address the above issues, with class re-balancing being the most popular heuristic. The two most prominent directions in the category include statistical *resampling* and *reweighting*. Resampling alters the exposure frequency during training, *i.e.*, more for the minorities and less so for the majorities [29]. Alternatively, reweighting directly amends the relative importance of each sample based on their class [22], sampling frequency [7] and associated cost for misclassification [30]. Recent developments also have considered class-sensitive and data-adaptive losses to more flexibly offset the imbalance [10, 60, 65, 55, 80, 51]. While being intuitive and working reasonably well in practice, an important is that these approaches offer no protection against the over-fitting of minority instances (see Figure 1)[91], a fundamental obstacle towards better generalization.

To circumvent this key limitation, inductive biases are often solicited to impose strong constraints that suppress spurious correlations that hinder out-of-sample generalization. A classical strategy is few-shot learning [31, 89, 40], where the majority examples are used to train meta-predictors and transferable features, leaving only a few parameters to tune for the minority data. In anomaly detection, methods such as one-class classification instead regard minority classes as outliers that do not always associate with stable, recognizable patterns [67, 75].

Despite their relatively strong assumptions, these methods capitalize on their superior ability in generalizing in the low sample regime in empirical settings.

More recently, establishing causal invariance has emerged as a new, powerful learning principle for better generalization ability even under apparent distribution shifts [73]. In contrast to standard empirical risk minimization (ERM) schemes, where the generalization to similar data distributions is considered, causally-inspired learning instead embraces robustness against potential perturbations [5]. This is achieved via only attending to causally relevant features and associations postulated to be invariant under different settings [70]. Specifically, contributions from spurious, unstable features are effectively blocked or attenuated. Interestingly, compromise of performance can be expected in those models [74], as a direct consequence of discarding useful (but non-causal) correlations in exchange for better causal generalization. Recently, [48] proposed non-parametric causal disentanglement of data representation to identify invariant relations.

This work explores the advancement of imbalanced data learning via adopting causal perspectives, with the insight that the key algorithm can be reformulated as an energy-based contrastive learning procedure to drastically improve efficiency and flexibility. In recognition of the limitations discussed above, we present *energy-based causal representation transfer* (ECRT): a novel imbalanced learning scheme that brings together ideas from causality, contrastive learning, energy modeling, data-augmentation and weakly-supervised learning, to address the identified weakness of existing solutions. Our key contributions are: ($i$) a causal representation encoder informed by an invariant generative mechanism based on generalized contrastive learning; ($ii$) integrated data-augmentation and source representation regularization techniques exploiting feature independence to enrich mi-

nority representations that better balance the trade-off between utility and invariance; $(iii)$ a key novelty is the derivation of an energy-based contrastive learning algorithm that greatly enhance model parallelism for large-label settings and extends generalized contrastive learning; and $(iv)$ insightful discussions on the justifications for the use of the proposed approach. Our claims are supported by strong experimental evidence.

## 2 Preliminaries

**Notation and problem definition.** We use $\boldsymbol{x} \in \mathbb{R}^p$ to denote the input data and $\boldsymbol{z} \in \mathbb{R}^d$ for the predictive features extracted from $\boldsymbol{x}$. Let $y \in \{1, \ldots, M\}$ be the class label. The number of training samples and those with label $m$ are denoted as $n$ and $n_m$, respectively. We use $\mathbb{E}[\cdot]$ to denote the expectation (average) of an empirical distribution, $\boldsymbol{a}_i$ to denote a vector associated with the $i$-th sample, and $[\boldsymbol{a}]_b$ to denote the $b$-th entry of vector $\boldsymbol{a}$. For simplicity, we assume class $M$ is the minority class, *i.e.*, $n_m \gg n_M$, for $m \in \{1, \ldots, M-1\}$. Throughout, we refer to $\mathcal{X}, \mathcal{Z}$ and $\mathcal{S}$ as data, feature, and source spaces, respectively, as shown in Figure 2, and defined below. Our goal is to accurately predict the label of minority instances with very limited training examples of it. Generalization to multiple minority categories is straightforward.

**Generalized contrastive learning and ICA** The proposed approach is based on a generalized form of *independent component analysis* (ICA), which addresses the inverse problem of signal *disentanglement* [46]. Specifically, ICA decorrelates features $\boldsymbol{Z}$ of the observed signal $\boldsymbol{X}$ into a source signal representation $\boldsymbol{S} = f_\psi(\boldsymbol{Z})$, where $f_\psi(\boldsymbol{z})$ is a smooth and invertible mapping known as the *de-mixing* function, and while assuming that the components of $\boldsymbol{S}$ are statistically independent, *i.e.*, with density $q(\boldsymbol{s}) = \prod_j q_j([\boldsymbol{s}]_j)$. Notationally, we call $[\boldsymbol{s}]_j$ the $j$-th *independent component* (IC) of $\boldsymbol{Z}$. While *nonlinear ICA* (NICA) is generally infeasible [19, 47], [48] has recently proposed a setting in which the identification of NICA can be achieved, by requiring an additional auxiliary label $y$. Specifically, NICA assumes that source signals are conditionally independent given $y$, *i.e.*, $q(\boldsymbol{s}|y) = \prod_j q_j([\boldsymbol{s}]_j|y)$, then $f_\psi(\boldsymbol{z})$ can be identified using *generalized contrastive learning* (GCL), whose implementation is detailed below.

GCL solves a generalized regression problem in which a *critic function* predicts whether label $y$ and representation $\boldsymbol{z}$ are correctly paired, *i.e.*, *congruent*. We call $(y_i, \boldsymbol{z}_i)$ a congruent pair and $(y_j, \boldsymbol{z}_i)$ an incongruent pair if $i \neq j$. Specifically, the critic is defined as $r_\nu(y, \boldsymbol{z}) = \sum_{a=1}^d r_\nu^a(y, [\boldsymbol{s}]_a)$, where $r_\nu^a(\cdot, \cdot)$ is a neural network with parameters $\nu$, whose inputs are the label $y$ and the $a$-th coordinate of $\boldsymbol{s} = f_\psi(\boldsymbol{z})$, denoted as $[\boldsymbol{s}]_a$. Then, GCL optimizes the following objective:

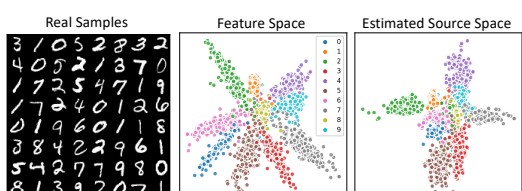

Figure 2: Illustration of data space $\mathcal{X}$, feature space $\mathcal{Z}$ (predictive but entangled) and source space $\mathcal{S}$ (independent, or disentangled) representations identified by ECRT for MNIST.

$$\underset{f_\psi, r_\nu}{\arg\min} \underbrace{\mathbb{E}_i[h(-r_\nu(y_i, \boldsymbol{z}_i))] + \mathbb{E}_{j \neq i}[h(r_\nu(y_j, \boldsymbol{z}_i))]}_{\mathcal{L}_{\mathrm{GCL}}(f_\psi, r_\nu)}, \tag{1}$$

where $h(r) = \log(1 + \exp(r))$ is the softplus function. (1) seeks to optimize $f_\psi(\cdot)$ and $r_\nu(\cdot, \cdot)$ by maximizing the discriminative power to tell apart congruent and incongruent pairs.

In fact, an interesting result showed by [48] revealed that maximizing the ability of the critic function $r_\nu(\cdot, \cdot)$ for correctly identifying matching pairs $(\boldsymbol{z}, y)$ leads to the identification (up to univariate transformation) of $f_\psi(\cdot)$, such that the components of $\boldsymbol{s}$ are conditionally independent given $y$. See the Supplementary Material (SM) for a formal exposition.

## 3 Energy-based Causal Representation Transfer

In this section, we describe the construction of *energy-based causal representation transfer* (ECRT): a causally informed data transformation and augmentation procedure to improve learning with imbalanced datasets. Our model assumes a shared causal data-generation procedure, which can be accurately identified by learning with the majority classes under assumed class-conditional representation independence. We obtain decorrelated representations that facilitate data augmentation and efficient learning with minority classes.

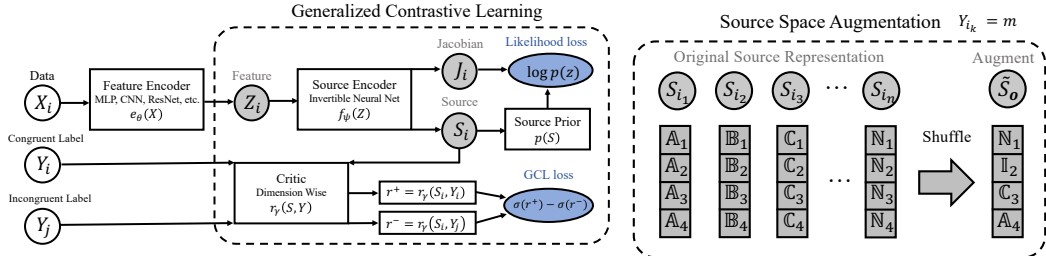

Figure 3: Source space estimation module of ECRT. We use GCL to identify the demixing function $s = f_\psi(z)$ via telling apart congruent & incongruent pairs.

Figure 4: Non-parametric source space augmentation based on shuffling.

The proposed model consists of the following components: $(i)$ a feature encoder module $z = e_\theta(x)$; $(ii)$ two classification modules, $h_{\phi'}(z)$ and $h_\phi(s)$, for predicting label $y$ from features $z$ and sources $s$, respectively; $(iii)$ a nonlinear ICA module for the de-mixing function $s = f_\psi(z)$; $(iv)$ a critic function $r_\nu(y, z)$ for GCL; and $(v)$ a data augmentation module. Further, $(\theta, \phi', \phi, \psi, \nu)$ denote the parameters of all the neural-network-based modules. Algorithm 1 outlines a general workflow, and below, we elaborate on our assumptions and detail the implementation of ECRT.

## 3.1 Model assumptions

To enable knowledge transfer across classes, we make the following assumptions:

**Assumption 3.1.** Let $z$ be a sufficient statistic (features) of $x$ for predicting label $y$, all class conditional feature distributions $p(z|y)$ share a common ICA de-mixing function $f_\psi(z)$.

---
**Algorithm 1** Energy-based Causal Representation Transfer.
1. Pre-train encoder and predictor:
$$e_\theta, h_{\phi'} \leftarrow \arg\min\{\mathbb{E}[\ell(h(e(x)), y)]\}$$
2. NICA estimation with fixed $z = e_\theta(x)$:
$$f_\psi, r_\nu \leftarrow \text{GCL}(\{(z, y)\}) \quad \% \text{ Equation (1)}$$
3. Source space augmentation with fixed $s = f_\psi(z)$:
$$(\tilde{s}, y) \leftarrow \text{AUG}(\{s|y = M\}) \quad \% \text{ Equation (2)}$$
4. Minority predictor modeling with augmented source:
$$h_\phi \leftarrow \arg\min\{\mathbb{E}[\ell(h(s, y = M)] + \lambda \mathbb{E}[\ell(h(\tilde{s}, y = M)]\}$$
---

This implies there exists a smooth invertible function $f_\psi : \mathcal{Z} \to \mathcal{S}$, and a set of IC distributions $\{q(s|y = m)\}_{m=1}^M$, that are linked to the conditional feature distributions $p(z|y = m)$ via $S^m = f_\psi(Z^m)$, where $S^m \sim q(s|y = m)$ and $Z^m \sim p(z|y = m)$. The subscript $m$ in $S^m$ and $Z^m$ indicates $y = m$, in a slight abuse of notation.

Importantly, $f_\psi^{-1}(s) : \mathcal{S} \to \mathcal{Z}$ is the *invariant causal mechanism* underlying the features of observed data. This specification, which is consistent with Assumption 3.1 enables the identification of the shared generating process, namely, the de-mixing function $f_\psi(z)$ that connects the likely dissimilar conditionals $\{p(z|y = m)\}_{m=1}^M$ via the source conditionals $\{q(s|y = m)\}_{m=1}^M$, as demonstrated in the context of NICA by [48]

A shortcoming of Assumption 3.1 is that the hypothesis supporting it is somewhat strong, untestable and may not hold in practice. However, we argue that the structural constraints imposed on the model by the assumption via the source conditionals $q(s|y = m)$ and the invertibility of $f_\psi(z)$, restrict the search space of the otherwise over-flexible model space powered by neural networks. Further, the causal mechanism implied by $f_\psi^{-1}(s)$ enables an effective knowledge transfer mechanism across classes via $q(s|y = m)$.

## 3.2 Energy-based Causal Representation Transfer

**Encoder pre-training.** To implement ECRT, we first find a good (reduced) feature representation of $x$ highly predictive of label $y$. This can be achieved via supervised representation learning (see Figure 3), which optimizes an encoder and predictor pair $(e_\theta(x), h_{\phi'}(z))$ to minimize the label prediction risk $\mathcal{L}(\phi') \triangleq \mathbb{E}[\ell(h_{\phi'}(e_\theta(x)), y)]$, where $\ell(\cdot, \cdot)$ is a suitable loss function, *e.g.*, cross-entropy, hinge loss, *etc.* To avoid capturing *spurious* (non-generalizable) features that overfit the minority class, we advocate training only with majority samples at this stage; assuming that $M > 2$. Alternatively, one could also consider unsupervised feature extraction schemes, such as auto-encoders. Further, we also recommend using statistical adjustments such as importance weighting, to reduce the impact of data imbalance.

**De-mixing representation with GCL.** After obtaining a good feature representation $\boldsymbol{Z} = e_\theta(\boldsymbol{X})$, we proceed to learn the de-mixing function $f_\psi(\boldsymbol{z})$, such that the coordinates of the source representation $\boldsymbol{S} = f_\psi(\boldsymbol{Z})$ are (approximately) independent given the label $y$. This can be done by optimizing the GCL objective in (1) with respect to the feature representation and label pairings $(y, \boldsymbol{z})$, adopting the *masked auto-regressive flow* (MAF) [69] to model the smooth, invertible transformation $f_\psi(\boldsymbol{z})$, which allows efficient parallelization of the autoregressive architecture via *causal masking* [14]. The procedure is outlined in Figure 3.

**Augmenting the minority.** Inspired by [83], we artificially augment the minority feature representations $\boldsymbol{z}$ via random permutations in the source space $\mathcal{S}$, as shown in Figure 4. Provided the assumed conditional independence of the sources, the features $\boldsymbol{Z}^m$ corresponding to label $y = m$ are generated by $\boldsymbol{Z}^m = f_\psi^{-1}(\boldsymbol{S}^m)$, where each dimension in the source representation $[\boldsymbol{S}^m]_j \sim q([\boldsymbol{s}]_j | y = m)$ are independently and *implicitly* sampled as described below. Specifically, using the (estimated) de-mixing function $f_\psi(\boldsymbol{z})$, we can obtain an approximate empirical source distribution for each $y = m$, *i.e.*, $S^m \triangleq \{\boldsymbol{s}_i = f_\psi(e_\theta(\boldsymbol{x}_i)) | y_i = m\} = \{\boldsymbol{s}_i\}_{i=1}^{n_m}$, where $S^m$ is a collection of $n_m$ samples of $q(\boldsymbol{s}|y = m)$. Then, we can draw new artificial samples $\tilde{\boldsymbol{s}}^m \sim q(\boldsymbol{s}|y = m)$ by randomly permuting the coordinates within elements $S^m$ independently via

$$\tilde{\boldsymbol{s}}_{\boldsymbol{o}}^m = ([\boldsymbol{s}_{o_1}^m]_1, [\boldsymbol{s}_{o_2}^m]_2, \cdots, [\boldsymbol{s}_{o_d}^m]_d), \tag{2}$$

where $\boldsymbol{o} = (o_1, \cdots, o_d)$ is a random permutation of $(1, \ldots, n_m)$. Note we have used $\tilde{\boldsymbol{s}}_{\boldsymbol{o}}^m$ to emphasize that the source point is artificially created via permutation $\boldsymbol{o}$. We call this procedure *nonparametric augmentation* because it does not make distributional assumptions for $q(\boldsymbol{s}|y = m)$. However, below we will discuss its limitations and consider an alternative where a parametric form is assumed. While it is tempting to refine the predictor $h_{\phi'}(\boldsymbol{z})$ with artificial features augmented via $\tilde{\boldsymbol{z}}^m = f_\psi^{-1}(\tilde{\boldsymbol{s}}^m)$, in Section 3.3 we will argue that it stands to benefit more from training a new predictor directly based on source representations, *i.e.*, $h_\phi(\boldsymbol{s})$, without the need for inverting $f_\psi(\boldsymbol{z})$.

**Model refinement.** Now we can leverage the augmented data to refine the prediction model. For minority class $y = M$, we optimize the following objective

$$\mathcal{L}_{\text{AUG}}(\phi') = \mathcal{L}(\phi') + \lambda(\mathbb{E}_{\tilde{\boldsymbol{Z}}^M}[\ell(h_{\phi'}(\tilde{\boldsymbol{z}}^M), M)] - \mathbb{E}_{\boldsymbol{Z}^M}[\ell(h_{\phi'}(\boldsymbol{z}), M)]), \tag{3}$$

where $\mathcal{L}(\phi')$ is the loss used for pre-training. Conceptually, (3) replaces a portion of the minority samples with augmentations. The trade-off parameter $\lambda \in [0, 1]$ encodes the relative confidence for trusting the artificially generated representations $\tilde{\boldsymbol{Z}}^M$ obtained from $S^M$ for the minority label $y = M$. Further, at this stage we found it's beneficial to fix the encoder module to prevent the de-mixing function to accommodate the changes in the encoder which in practice may cause instability during training.

**Challenges with naïve implementation.** We identify three major issues with naïvely implemented ECRT, to be addressed in the section below: ($i$) *Representation conflict*: since the GCL solution is not unique, we do observe naïve GCL training drifts among viable source representations whose performance differ considerably, causing stability concerns; ($ii$) *Costly augmentation*: MAF inversions dominate the computation load during training, which becomes prohibitive in high dimensions; and ($iii$) *Gridding artifact*: a small minority sample size leads to pronounced augmentation bias when sampling nonparametrically via (2), manifested as a rectangular-shaped grid (see Figure 5).

### 3.3 Improving causal representation transfer

**Energy-based GCL.** Our key insight to improve GCL comes from the fact that Equation (1) is essentially learning the density ratio between the joint and product of marginals, *i.e.*, $\frac{p(x,y)}{p(x)p(y)}$. This immediately reminds us the recent literature on contrastive *mutual information* (MI) estimators, such as InfoNCE [71]. In such works, a variational lower bound of MI is derived, and the algorithm optimizes a critic function using the positive samples from the joint distribution, and the negative samples from the product of the marginals. At their optimal value, these critics recover the density ratio or a transformation of it. Our development is based on the recent work of [37], using an energy-perspective to improve contrastive learning. Specifically, we will be using a variant of the celebrated *Donsker-Varadhan* (DV) estimator [28], and applied Fenchel duality trick to compute a solution [32, 81, 23]. Specifically, the *Fenchel-Donsker-Varadhan* (FDV) estimator takes the following form:

$$I_{\text{FDV}} \triangleq \hat{I}_{\text{DV}}^K(\{\boldsymbol{x}_i, \boldsymbol{y}_i\}) + \frac{\sum_j \exp[(g_\theta(\boldsymbol{x}_i, \boldsymbol{y}_j) - g_\theta(\boldsymbol{x}_i, \boldsymbol{y}_i))/\tau]}{\sum_j \exp[(\hat{g}_\theta(\boldsymbol{x}_i, \boldsymbol{y}_j) - \hat{g}_\theta(\boldsymbol{x}_i, \boldsymbol{y}_i))/\tau]} + 1, \tag{4}$$

where $g_\theta(\boldsymbol{x}_i, \boldsymbol{y}_i)$ is our critic of interest and $\hat{I}_{\text{DV}}^K(\{\boldsymbol{x}_k, \boldsymbol{y}_k\}) = g(\boldsymbol{x}_1, \boldsymbol{y}_1) - \log(\sum_{k'=1}^K \exp(g(\boldsymbol{x}_1, \boldsymbol{y}'_k))/K)$ is the *Donsker-Varadhan* (DV) estimator [28] for the MI. Using the same parameterization used in GCL recovers the same causal identification property (see Appendix for details). Compared to the original GCL formulation, we are now using multiple negative samples instead of one, which greatly boosts learning efficiency [38]. And this can be efficiently implemented with the *bilinear* critic trick [15, 13, 37] so that all in-batch data can be used as negatives. In our context, it greatly boosts training efficiency when dealing with a large number of different classes. See Algorithm S1 in Appendix.

**Regularizing the data likelihood.** Recall GCL solutions can only be identified up to an invertible transformation of each dimension [48], and the predictive performance of different valid GCL solutions can vary significantly. Empirically, we observe that a naïve implementation of GCL often leads to source representations that are densely packed (see Figure S1 in the SM). This is undesirable, when decoding back to the feature space and making useful predictions, the neural network predictor will need to be expansive, *i.e.*, requiring a large Lipschitz constant, thereby sacrificing optimization stability and model generalization according to existing learning theory [20, 87].

To encourage source representations that are less condensed, we consider a simple, intuitive strategy consisting of regularizing the source representation with the $\log$-likelihood in feature space. This likelihood can be easily obtained with MAF using $\ell_{\text{FLOW}}(f_\psi)$ defined in the SM. Following common practice, we set source prior $p(\boldsymbol{s})$ to the standard Gaussian, and optimize the following likelihood-regularized GCL objective:

$$\tilde{\mathcal{L}}_{\text{GCL}}(f_\psi, r_\nu) = \mathcal{L}_{\text{GCL}}(f_\psi, r_\nu) + \rho \mathcal{L}_{\text{FLOW}}(f_\psi), \tag{5}$$

where $\rho > 0$ is the regularization strength. Naturally, this regularization will encourage the global source representation identified by GCL to be more consistent with a Gaussian-shaped distribution.

An alternative interpretation for the likelihood-regularized objective in (5) is that it can be understood as a relaxation to the conditional independence (Assumption 3.1). To see this, recall the likelihood objective obtained from the *invertible neural networks* (INN) alone attempts to map the source representations to be *unconditionally* independent, as opposed to the *conditional* independence assumed by

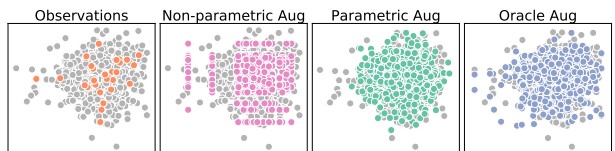

Figure 5: Comparison of different source augmentations overlaid on the ground-truth distribution (gray dots). A severe *gridding artifact* is observed in the low-sample regime for the nonparametric scheme, whereas the parametric augmentation closely matches the oracle in distribution.

NICA. The regularized formulation (5) provides a safe *"fall-back"* mode in case Assumption 3.1 is violated. This also motivates us to consider an important variant: making multiple class-dependent source priors, *i.e.*, $p^m(\boldsymbol{s})$ for each class label $m \in \{1, \cdots, M\}$ in (5), whose parameters (*i.e.*, mean and variance) are jointly learned with other model parameters. Compared to the fully non-parametric objective (1), this strategy further encourages the source representations to be independent given the class labels, and it enables *parametric* data augmentation, *i.e.*, sampling from the parametric label priors instead of permuting the indices. We refer to this variant as ECRT-MP, where MP stands for multiple priors. And similarly, ECRT-1P refers to the case when a single prior is used. We have found that ECRT-MP performs better in most cases, and consequently, ECRT means ECRT-MP by default.

**Modeling in the source space.** Rather than modeling the predictor $h_{\phi'}(\boldsymbol{z})$ in the feature space $\mathcal{Z}$, we advocate instead for building the predictor directly in the source space $\mathcal{S}$, *i.e.*, modeling with $h_\phi(\boldsymbol{s})$. This practice enjoys several benefits: ($i$) *Easy & robust augmentation*: many designs of high-dimensional flows are asymmetric computationally, and inverting a MAF is not only $d$ times more costly than a forward pass, it is also numerically unstable at the boundary. Direct modeling in the source space circumvents the difficulties associated with MAF inversions during data augmentation; ($ii$) *Feature whitening*: the source representation identified by GCL is component-wise independent, and literature documents abundant empirical evidence that similar de-correlation based pre-processing, commonly known as *whitening*, benefits learning [45, 6, 49].

**Parametric augmentation.** When the number of minority observations is scarce, the above non-parametric indices-shuffling augmentation suffers from the *gridding artifact* (Figure 5). This artifact amplifies the augmentation bias in the low-sample regime. To overcome this limitation, we empirically observe that the estimated class-conditional source distributions are usually Gaussian-like after the

likelihood regularization (especially so when label conditional priors are used). In these situations, a parametric augmentation that draws synthetic source samples from a Gaussian distribution matched to the empirical mean and variance of minority source representations is more efficient.

### 3.4 Insights and remarks

To better appreciate the gains and limitations expected from ECRT, we compile a few complementary arguments below, through the lens of very different perspectives.

**Why causal augmentation works.** It is helpful to understand the gains from ECRT's causal augmentation beyond the heuristic that permuting the ICs provides more training samples for the minority class. [83] considered a similar causal augmentation procedure for few-shot learning, and provided two major theoretical arguments: $(i)$ the risk estimator based on the augmented examples is the uniformly minimum variance unbiased estimator given the accurate estimation of $f_\psi$ (see Theorem 1, [83]); and $(ii)$ with high probability, the generalization gap can be bounded by the approximation error of $f_\psi$ (see Theorem 2, [83]). In the SM, we give arguments that our causal augments give the 'best' label-conditional distribution estimate.

**Speedup from shared embedding.** While for typical supervised learning tasks the generalization bound scale as $\mathcal{O}(n^{-\frac{1}{2}})$, a superior rate of $\mathcal{O}(n^{-\eta})$ where $\eta \in [\frac{1}{2}, 1]$ is possible, if there exists abundant data for an alternative, yet related task that shares the same feature embedding (see Theorem 3, [72]). Note $n$ refers to the size of labeled data directly related to the *strong* task of interest, in our case, prediction of minority labels. Our ECRT employs GCL to identify one such common embedding, *i.e.*, the source space, using the majority examples, and consequently, improves predictions on the minority class.

**Representation whitening.** Our ECRT causally disentangles representation [78, 84] via de-correlating the representations conditionally. Extensive empirical evidence has pointed to the fact that such representation de-correlation, more commonly known as data whitening [52], is expected to considerably improve learning efficiency [18]. This benefit has been attributed to the better conditioning of the Fisher information matrix for gradient-based optimization [25], thus rectifying second-order curvatures to accelerate convergence. Our source space modeling explicitly separates the task of representation disentanglement, and in turn, helps the prediction network to focus on its primary goal.

**Potential limitations.** The setting considered by ECRT is restrictive in that it precludes the learning of useful, yet non-transferable features predictive of the minority labels. For instance, there might be a feature unique to the minority class. However, since the de-mixer $f_\psi(z)$ is only trained on the majority domains absent of this feature, it can not be accounted for by the ECRT model. This is a key limitation of causally inspired models, in that they are often too conservative for only retaining the invariant features, promoting cross-domain generalization at the cost of within-domain performance degradation [74].

## 4 Related Work

**Causal invariance and representation learning.** A major school of considerations for building robust machine learning models is to stipulate invariant causal relations across environments [76, 62, 12], such that one hopes to safely extrapolate beyond the training scenario [8]. We broadly categorize such efforts into two streams, namely predictive causal models and generative causal models. Prominent examples from the first category include ICP [70, 43] and IRM [5], which highlight the identification of invariant representation and causal relations via penalizing environmental heterogeneity. Our solution pertains to the second category, where the data distribution shifts across environments can be tethered by an invariant generation procedure [48, 53]. This work exploits the recovered causally invariant source representation to improve learning efficiency and mitigate the sample scarcity of minority labels in imbalanced sets.

**Learning with imbalanced data.** Resolving data imbalance is a heavily investigated topic [61]. Standard sampling and weight adjustments suffer from caveats such as introducing bias and information loss. Due to these concerns, recent literature has actively explored adaptive strategies, such as redundancy-adjusted balancing weights [22], and the theoretically grounded class-size adapted margins[10]. Similar to *boosting* [34], adaptive weights are designed to prioritize the learning of less well-classified examples [60] while excluding apparent outliers [59]. Much related to our setting are the *meta-learning* scenarios [88, 33, 90], where the model tries to generalize & repurpose the

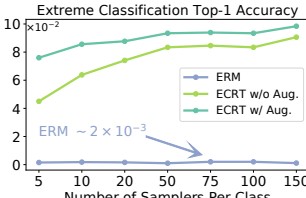
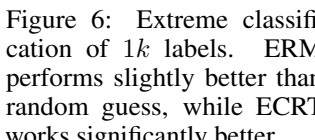

Figure 6: Extreme classification of $1k$ labels. ERM performs slightly better than random guess, while ECRT works significantly better.

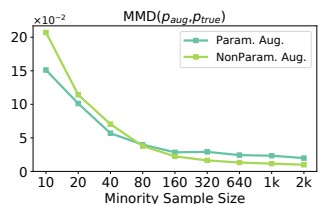

Figure 7: Comparison of different causal augmentations, lower is better. Parametric augmentation is more efficient with small samples.

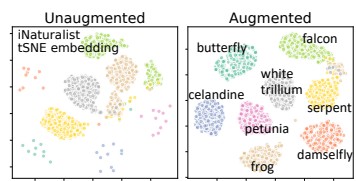

Figure 8: ECRT source representation trained on `iNaturalist`, visualized using tSNE embedding in two dimensions for eight random categories.

knowledge learned from majority classes to minority predictions; and also the augmentation-based schemes to restore class balance [64, 68].

**Data augmentation.** Due to its exceptional effectiveness, augmentation schemes are widely adopted in practical applications [77]. These augmentation strategies are built on known invariant transformations, *e.g.*, rotation, scaling, noise corruption, *etc*. [79], simple interpolation heuristics [11, 41], and more recently towards fully auto-mated procedures [21]. Notably, recent trend in data augmentation highlights robust learning against adversarially crafted inputs [35] and generative augmentation procedures that compose realistic artificial samples [3].

**Domain adaption and causal mechanism transfer.** Closest to our contribution is *causal mechanism transfer* (CMT) [83], which focused on addressing few-shot learning for continuous regression. We note a few key differences to our work: $(i)$ CMT focused on domain adaptation and does not address classification; and $(ii)$ it bundles $(\boldsymbol{x}, y)$ for NICA which necessitates the flow inversion for sample augmentation. Grounded in the setting of imbalanced data learning, our ECRT extends applications and advocates source space modeling to simplify and improve causal augmentation. It also features likelihood regularization to enhance representation regularity. We also offer new insights to justify the use of NICA-based augmentation, complementing the analysis from CMT by [83].

**Energy-based modeling for representation learning.** There is growing recognition that the energy perspective is integral to representation learning [58, 71, 82, 37]. In the lens of energy based modeling, the distribution of data is characterized as an (unnormalized) energy function [4]. Optimization of the energy function is often considered challenging [81] and contrastive techniques have been proven effective [44, 39]. Our work is a generalization of [48] that disentangles representations from an energy perspective. Interesting comparison can be made to [54], whose training objective bears resemblance to our ECRT. But the specific designs used by ECRT in the network architecture and scoring function allows ECRT to provably disentangle the feature representations and consequently capture causality for exploitation.

## 5 Experiments

To validate the utility of our model, we consider a wide range of (semi)-synthetic and real-world tasks experimentally. All experiments are implemented with PyTorch, and our code is available from `https://github.com/ZidiXiu/ECRT`. More details of our setup & additional analyses are deferred to the SM Sections C-E.

### 5.1 Experimental setup

**Baselines.** The following competing baselines are considered to benchmark the proposed solution: $(i)$ Empirical risk minimization (ERM), a naïve baseline with no adjustment; $(ii)$ Importance-weighting (IW) [9], a class-weight balanced training loss; $(iii)$ Generative adversarial augmentation (GAN) [3], synthetic augmentations from adversarially-trained sampler; $(iv)$ Virtual adversarial training (VAT) [66], robustness regularization with virtual perturbations; $(v)$ FOCAL loss [60], cost-sensitive adaptive weighting; $(vi)$ Label-distribution-aware margin loss (LDAM) [10], margin-optimal class weights. All baselines are tuned for best performance for TOP-1 accuracy on validation.

**Evaluation metrics & setup.** We consider the following metrics to quantitatively assess performance: $(i)$ *negative* log-*likelihood* (NLL); $(ii)$ F1 score; $(iii)$ TOP-$k$ accuracy ($k = 1, 5$). Following the classical evaluation setup for imbalanced data learning, we learn on an imbalanced training set and report performance on a balanced validation set.

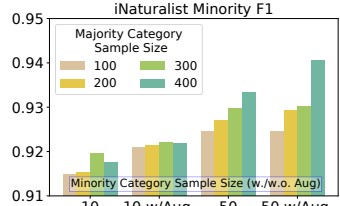
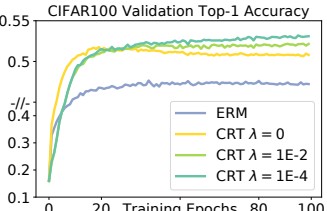
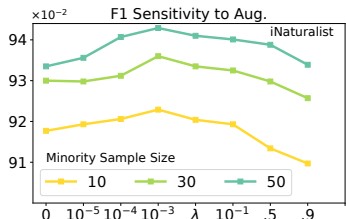

Figure 9: Performance comparison with and without augmentation. Augmented solutions improve the effective sample size.

Figure 10: Comparison of learning dynamics. ECRT enables both faster learning and better model predictions.

Figure 11: Sensitivity analysis of augmentation strength $\lambda$. Smaller minority sample sizes are more sensitive to the augmentation.

Table 1: Comparison of performance on real-world datasets ($\uparrow$ higher is better, $\downarrow$ lower is better).

|          | CIFAR100 | | | iNaturalist | | | TinyImageNet | | | ArXiv | |
|          | Top-1$\uparrow$ | Top-5$\uparrow$ | NLL$\downarrow$ | Top-1$\uparrow$ | Top-5$\uparrow$ | NLL$\downarrow$ | Top-1$\uparrow$ | Top-5$\uparrow$ | NLL$\downarrow$ | Acc$\uparrow$ | NLL$\downarrow$ |
|----------|---------|---------|---------|---------|---------|---------|---------|---------|---------|---------|---------|
| ERM      | 49.29 | 78.22 | 2.95 | 66.73 | 87.86 | 1.70 | 58.52 | 79.01 | 3.22 | 44.64 | **0.0407** |
| IW       | 43.97 | 68.96 | 3.89 | 67.63 | 88.94 | 1.66 | 60.50 | 80.23 | 2.92 | 46.02 | 0.0477 |
| GAN      | 47.64 | 78.53 | 2.69 | 67.40 | 87.00 | 1.82 | 60.69 | 80.91 | 2.33 | 45.42 | **0.0407** |
| VAT      | 46.47 | 74.23 | 3.19 | 67.06 | 87.39 | 1.90 | 59.69 | 82.05 | 2.42 | 45.82 | 0.0474 |
| FOCAL    | 43.32 | 74.39 | 2.89 | 66.63 | 88.45 | 1.59 | 58.27 | 79.39 | 2.59 | 46.01 | 0.0416 |
| LDAM     | 50.46 | 74.39 | 2.18 | 67.39 | 87.13 | 4.00 | 58.18 | 82.51 | 2.15 | 45.04 | 0.0450 |
| ECRT-1P  | 52.31 | 81.14 | **1.98** | 68.38 | 88.13 | 1.48 | 62.46 | 83.50 | **1.79** | 48.33 | 0.0434 |
| ECRT-MP  | **53.00** | **81.99** | 2.31 | **69.01** | **90.01** | **1.23** | **64.40** | **84.54** | 1.94 | | |

**Toy model.** We sample 2D standard Gaussians with different means and variances as source representation $s$ for each label class, which is then distorted by random affine and Hénon transformations $z_1 = 1 - 1.4\,\tilde{s}_1^2 + \tilde{s}_2, z_2 = 0.3\,\tilde{s}_1$, to induce real-world-like complex association structures. See the SM Sec. D for a detailed description and visualizations.

**Real-world datasets.** We consider the following semi-synthetic and real datasets: ($i$) Imbalanced `MNIST` and `CIFAR100`: standard image classification tasks with artificially created step-imbalanced following [10]; ($ii$) Imbalanced `TinyImageNet` [2]: a scaled-down version of the classic natural image dataset `ImageNet`, comprised of 200 classes, 500 samples per class and $10k$ validation images, with different simulated imbalances applied; ($iii$) `iNaturalist` 2019 [86]: a challenging task for image classification in the wild comprised of $26k$ training and $3k$ validation images labeled with $1k$ classes, with a skewed distribution in label frequency; ($iv$) `arXiv` abstracts, imbalanced multi-label prediction of paper categories with $160k$ samples and 152 classes.

**Preprocessing, model architecture, and tuning.** We use pre-trained models to extract vectorized representations for big complex datasets: ResNet [42] for image models of `TinyImageNet`[2], Inception V3 for `iNaturalist`[3], and BERT [26][4] for the `arXiv` [1] language model. These pre-trained representations are used as raw feature inputs, subsequently fed into a fully-connected multi-layer perceptrons (MLP) for source space encoding. For small image models (*i.e.*, `MNIST`, `CIFAR100`) we directly train a CNN from scratch for feature extraction. We used a random $8/2$ split for training and validation, and applied Adam optimizers for training. We rely on the best out-of-sample cross-entropy and GCL loss for hyperparameter tuning.

**ECRT efficiency.** We first examine the dynamics of efficiency gains from ECRT using a toy model, with the results summarized in Figure 9. Consistent with the shared embedding perspective, increasing the majority sample size improves the minority accuracy. Causal augmentation consistently improves performance, and it is most effective in the low-sample regime, offering more than $4\times$ boosts in effective minority sample size. In Figure 10, we compare modeling with feature and source representations, respectively. Source space modeling speeds up learning, in addition to the modest accuracy gains. Augmented training is slightly slower, but eventually converges to a better solution.

[2]`https://download.pytorch.org/models/resnet18-5c106cde.pth`
[3]`https://download.pytorch.org/models/inception_v3_google-1a9a5a14.pth`
[4]`https://github.com/allenai/scibert`

Interestingly, the major performance gain originates from the GCL-based source space modeling we proposed, and to a lesser extent, from the augmentation perspective presented by [83].

**Ablations on augmentation.** We examined the contributions from augmentation as we vary the augmentation strength $\lambda$ (Figure 11). A relatively small $\lambda$ already yields improvement, but when the minority size is small, stronger augmentation degrades performance. This implies the augmentation gains are more likely to be originated from the exposure to the diversity of synthetic augmenting, and the accuracy of augmented distribution is limited by the imperfectness of empirical estimation (*e.g.*, limited sample size). In Figure 7 we compare the MMD distance [36] between the parametric and nonparametric augmentation outputs and the ground-truth, confirming the improved efficiency from the ECRT parametric augmentation in the low-sample regime.

**Ablation on different model variants.** We further compare the model performance with feature encoder trained with both majority and minority. (Table 1 used feature encoder trained only majority classes.) We have `Cifar100` TOP-1=51.79, TOP-5=81.02, NLL=2.03. This is slightly worse than the majority-only result but still outperformed other competing solutions. We caution, whether including minority examples in the pre-training of feature encoder differs case-by-case. They may not affect performance at all (majority dominant), improve performance (predictive features consistent with those used by majority) and completely devastating (containing spurious features, overfit).

**Extreme classification.** A related challenge of interest is extreme classification [16], where the label size is huge but there is only a handful of samples pertaining to each label class, This means there is no clear majority class. In Figure 6, we show the proposed ECRT also fares very well in this scenario, while standard ERM struggles (slightly better than random guess). This result evidences that ECRT also efficiently extracts generalizable information from an abundance of different class labels.

**Imbalanced data learning.** In Table 1 we compare the performance of ECRT to other competing solutions, with each baseline carefully tuned to ensure fairness. Minor discrepancies compared to results reported by prior literature are explained in the SM Sec. E, and in general our implementation performs slightly better. Overall, ECRT-based solutions consistently lead the performance chart, with the label-prior variant solidly outperforming vanilla ECRT and other competitors in most categories. While showing varying degrees of success, FOCAL and LDAM failed to establish dominance compared to ERM. GAN and VAT also verified the effectiveness of augmentation and adversarial perturbations in the imbalanced setting. We also compare the overall F1-score between different methods on `CIFAR100` dataset: ERM: 0.439, SMOTE [11]: 0.444, FOCAL:0.391, LDAM: 0.408, and ECRT: 0.482. FOCAL and LDAM gave very poor F1 scores in this case, even worse than the ERM baseline. SMOTE improved ERM, but the largest gain is obtained by ECRT. Figure 1 compares performance of different learning schemes condition on different label sizes using the F1 score on the `iNaturalist` data. In Figure 8 we show the ECRT learned representations, with and without augmentation, using tSNE embeddings.

## 6  Conclusion

This paper developed a novel learning scheme for imbalanced data learning. Leveraging a causal perspective, our solution cuts through the apparent data heterogeneity and identifies a shared, invariant, disentangled source representation, using only majority samples. We demonstrate this source representation modeling enables more efficient learning and allows principled augmentation. Importantly, we bridge the research between contrastive representation learning and energy modeling to causal machine learning, which constructs promising directions for future research.

## Acknowledgements

This research was supported in part by NIH/NIDDK R01-DK123062, NIH/NIBIB R01-EB025020, NIH/NINDS 1R61NS120246, DARPA, DOE, ONR and NSF. J. Chen was partially supported by Shanghai Municipal Science and Technology Major Project (No.2018SHZDZX01) and National Key R&D Program of China (No.2018AAA0100303). This work used the Extreme Science and Engineering Discovery Environment (XSEDE), which is supported by National Science Foundation grant number ACI-1548562 [85]. This work used the Extreme Science and Engineering Discovery Environment (XSEDE) PSC Bridges-2 and SDSC Expanse at the service-provider through allocation TG-ELE200002 and TG-CIS210044.

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
