# Supercharging Imbalanced Data Learning With Energy-based Contrastive Representation Transfer (Supplementary Material)

Junya Chen, Zidi Xiu, Benjamin Goldstein, Ricardo Henao, Lawrence Carin, Chenyang Tao
Duke University

## Contents

## A. Theoretical Support

Here we summarize some theories from literature that supports the development of this paper by making it self-contained. Attempts have been made to unify the notations, making them consistent with our paper, and also drop some contents from the original presentations that not directly relevant in this context.

### A.1. Nonlinear ICA with auxiliary variables

The following theory lists the technical conditions required for the identification of conditional nonlinear ICA model, based on which our work was built.

**Definition A.1** (Conditionally exponential of order $k$). A random variable (independent component) $[\boldsymbol{s}]_i$ is conditionally exponential of order $k$ given random vector $\boldsymbol{c}$ if its conditional pdf can be given in the form

$$p([\boldsymbol{s}]_i | \boldsymbol{c}) = \frac{Q_i([\boldsymbol{s}]_i)}{Z_i(\boldsymbol{c})} \exp \left[ \sum_{j=1}^{k} \tilde{q}_{ij}([\boldsymbol{s}]_i) \lambda_{ij}(\boldsymbol{c}) \right] \tag{S1}$$

almost everywhere in the support of $\boldsymbol{c}$, with $\tilde{q}_{ij}, \lambda_{ij}, Q_i$ and $Z_i$ scalar-valued functions. The sufficient statistics $\tilde{q}_{ij}$ are assumed linearly independent.

**Theorem A.2** (Theorem 3, [4], identification of Nonlinear ICA). *Assume (i) the data follows the nonlinear ICA model with the conditional independence $q(\boldsymbol{s}|\boldsymbol{c}) = \prod_j q_j([\boldsymbol{s}]_j|\boldsymbol{c})$; (ii) Each $[\boldsymbol{s}]_j$ is conditionally exponential given $\boldsymbol{c}$ (Definition A.1); (iii) There exist $nk + 1$ points $\boldsymbol{c}_0, \cdots, \boldsymbol{c}_{nk}$, such that the following matrix of size $nk \times nk$*

$$\tilde{\boldsymbol{L}} = \left( \begin{array}{ccc} \lambda_{11}(\boldsymbol{c}_1) - \lambda_{11}(\boldsymbol{c}_0) & \cdots & \lambda_{11}(\boldsymbol{c}_{nk}) - \lambda_{11}(\boldsymbol{c}_0) \\ \lambda_{nk}(\boldsymbol{c}_1) - \lambda_{nk}(\boldsymbol{c}_0) & \cdots & \lambda_{nk}(\boldsymbol{c}_{nk}) - \lambda_{nk}(\boldsymbol{c}_0) \end{array} \right) \tag{S2}$$

*is invertible; (iv) nonlinear Logistic regression system Eqn (1) is trained using functions with universal approximation capacity. Then in the limit of infinite data, $f(\boldsymbol{z})$ provides a consistent estimator of the nonlinear ICA model, up to a linear transformation of point-wise scalar functions of the independent components.*

## A.2. Variance and generalization bound

The following theories explore the consequence of training using only nonparametric causal augmentation. First we define the risk estimators.

**Definition A.3.** Let $\tilde{S}$ be the non-parametric source augmentation defined in Eqn (3) main text, $\ell(\cdot)$ be the loss function, $g(\boldsymbol{z})$ be the hypothesis function. We define the risk $R$ and causally augmented risk estimator $\breve{R}$ wrt $g$ respectively as

$$R(g) \triangleq \mathbb{E}_Z[\ell(g(Z))], \breve{R}(g) \triangleq \mathbb{E}_{\tilde{S}}[\ell(g(\hat{f}^{-1}(\tilde{S})))], \tag{S3}$$

where $\hat{f}$ is the estimated causal de-mixing function.

The following theorem revealed that assuming perfect knowledge of de-mixing function $f$, the causally augmented risk estimator is optimal.

**Proposition A.4.** Assuming $\hat{f} = f$, and let $\ell(h(\boldsymbol{x}), y)$ be the classification loss for predictor $h \in \mathcal{H}$. Let $\hat{R}(h) = \sum_m w_m \hat{R}_m(h)$ be an estimator for $R(h) = \mathbb{E}[\ell(h(\boldsymbol{x}), y)]$, such that $\hat{R}(h)$ is an unbiased estimator for $R_m(h) = \mathbb{E}_{y=m}[\ell(h(\boldsymbol{x}), y)]$. Then $\tilde{R}(h) = \sum_m w_m \tilde{R}_m(h)$, where $\tilde{R}_m(h) \triangleq \mathbb{E}_{y_i=m}[\ell(h(\tilde{\boldsymbol{x}}_i), y_i)]$ is the minimum variance unbiased estimator among all $\hat{R}(h)$.

*Proof.* This is a direct consequence of Theorem A.5. If $\tilde{R}(h)$ is not the minimal variance estimator, then at least one of $\tilde{R}_m(h)$ is not optimal, which contradicts Theorem A.5. $\qquad \square$

**Theorem A.5** (Theorem 1, [9], minimum variance property). *Assuming $\hat{f} = f$. Then for each $g \in \mathcal{G}$, the causal augmented risk estimator $\tilde{R}(g)$ is the uniformly minimum variance unbiased estimator of $R(g)$, i.e., $\mathbb{E}[\tilde{R}(g)] = R(g)$ and for any unbiased estimator $\breve{R}$ of $R(g)$ (i.e., $\mathbb{E}[\breve{R}(g)] = R(g)$),*

$$Var[\tilde{R}(g)] \leq Var[\breve{R}(g)]. \tag{S4}$$

Since we are bound to have estimation errors, the next theorem establishes the generalization bounds wrt such errors.

**Theorem A.6** (Theorem 2, [9], excess risk bound). *Let $\breve{g} = \arg\min \breve{R}$ and $g^* = \arg\min R(g)$, then under appropriate assumptions (Assumptionss 1-8 in [9]), for arbitrary $\delta, \delta' \in (0, 1)$, we have probability at least $1 - (\delta + \delta')$,*

$$R(\breve{g}) - R(g^*) \leq \underbrace{C \sum_{j=1}^{d} \|f_j - \hat{f}\|_{W^{1,1}}}_{Approximation\ error} + \underbrace{4d\mathfrak{R}(\mathcal{G}) + 2dB_\ell \sqrt{\frac{\log 2/\delta}{2n}}}_{Estimation\ error} + \underbrace{\kappa_1(\delta', n) + dB_\ell B_q \kappa_2(f - \hat{f})}_{Higher\text{-}order\ terms}. \tag{S5}$$

Here $\|\cdot\|_{W^{1,1}}$ is the Sobolev norm and $\mathfrak{R}(\mathcal{G})$ is the effective Rademacher complexity defined by

$$\mathfrak{R}(\mathcal{G}) \triangleq \frac{1}{n} \mathbb{E}_{\hat{S}} \mathbb{E}_\sigma \left[ \sup_{g \in \mathcal{G}} \left| \sum_{i=1}^{n} \sigma_i s \mathbb{E}_{S_2', \cdots, S_d'} \tilde{\ell}(\hat{s}_i, S_2', \cdots, S_d') \right| \right], \tag{S6}$$

where $\{\sigma_i\}_{i=1}^n$ are independent sign variables, $\mathbb{E}_{\hat{S}}$ is the expectation wrt $\{\hat{s}_i\}_{i=1}^n$, the dummy variables $S_2', \cdots, S_d'$ are i.i.d. copies of $\hat{s}_1$, and $\tilde{\ell}$ is defined by

$$\tilde{\ell}(s_1, \cdots, s_d) \triangleq \frac{1}{d!} \sum_\pi \ell(g, \hat{f}^{-1}(s_{\pi(1)}, \cdots, s_{\pi(d)})), \tag{S7}$$

where $\pi$ denotes the permutations. $\kappa_1, \kappa_2$ are higher order terms, $B_q, B_\ell$ respectively depends on density $q$ and loss $\ell$, while $C'$ depends on $(f, q, \ell, d)$.

## A.3. Speedup from shared embedding

[8] built some interesting theories trying to answer the following question: "*Can large amounts of weakly labeled data provably help learn a better model than strong labels alone?*" The answer is positive, assuming there is a shared embedding between the *weak* and *strong* tasks, which respectively refers auxiliary (secondary) and main tasks of interests. We summarize its main findings below and elaborate how it lends support for ECRT.

In the setting of weakly supervised learning, we have the triplet $(\mathcal{X}, \mathcal{W}, \mathcal{Y})$, where $\mathcal{X}$ and $\mathcal{Y}$ respectively denote the features and labels of interest (strong task), and $\mathcal{W}$ denote weak task labels that are relevant to the prediction of $\mathcal{Y}$. It is assumed that there is this unknown good embedding $Z = f_0(X)$ that predicts $W$, that could be leveraged to derive a model of the form $\hat{g}(\cdot, \hat{f}) : \mathcal{X} \to \mathcal{Y}$ that improves learning.

---

**Algorithm 1** Weakly supervised learning

---

1. Pretrain encoder with weak labels

$$\hat{f} \leftarrow \text{Alg}(\mathcal{F}, \mathbb{P}_{XW})$$

2. Augment data with

$$z_i = \hat{f}(x_i) \Rightarrow \{(x_i, y_i, z_i)\} \sim \hat{\mathbb{P}}_{XYZ}$$

3. Optimize the strong task

$$\hat{g} \leftarrow \text{Alg}(\mathcal{G}, \hat{\mathbb{P}}_{XYZ})$$

---

**Theorem A.7** (Theorem 3, [8]). *Suppose that* $Rate_m(\mathcal{F}, \mathbb{P}_X W) = \mathcal{O}(m^{-\alpha})$ *and that* $Alg_n(\mathcal{G}, \hat{\mathbb{P}})$ *is ERM. Under suitable assumptions on* $(\ell, \mathbb{P}, \mathcal{F})$, *Algorithm 1 obtains excess risk*

$$\mathcal{O}\left(\frac{\alpha\beta \log n + \log(1/\delta)}{n} + \frac{1}{n^{\alpha\beta}}\right) \tag{S8}$$

*with probability* $1 - \delta$, *when* $m = \Omega(n^\beta)$ *for* $\mathcal{W}$ *discrete, or* $m = \Omega(n^{2\beta})$ *for* $\mathcal{W}$ *continuous.*

For concrete examples, in a typical learning scenario where $\text{Alg}_m(\mathcal{F}, \mathbb{P}_{XW}) = \mathcal{O}(m^{-1/2})$, one obtains the fast rates $\mathcal{O}(1/n)$ for $m = \Omega(n^2)$.

In the context of our ECRT, we identify the learning of common causal de-mixing function $f(z)$ as the weak learning task, and the source space $\mathcal{S}$ is the common embedding space of interest. This allows us to tape into the power of weakly supervised learning to improve the main classification task. See Figure 9 in the main text for evidence.

## A.4. Energy-based GCL

Mutual information (MI) is a popular metric to quantify the associations between random variables, and has been applied to a lot of areas like independent component analysis, fair learning and etc. Inspired by the FDV loss introduced in [3], rooted from the MI between the feature $z$ and label $y$. Following the Equation (4) in the main text for $I_{\text{FDV}}$, we have a novel MI objective, pointwise mutual information (PMI) which takes the place of logistic regression objective in GCL,

$$\mathcal{L}_{\text{FDV}}(f_\psi, g_\nu) \triangleq \hat{I}_{\text{DV}}(\{z_i, y_i\}) + \frac{\sum_j \exp[(g_\nu(y_j, z_i) - g_\nu(y_i, z_i))/\tau]}{\sum_j \exp[(\hat{g}_\nu(y_j, z_i) - \hat{g}_\nu(y_i, z_i))/\tau]} - 1, \tag{S9}$$

where $y_i$ is the embedding of label, $\tau$ is a learnable temperature parameter, $g_\nu(y, z) = \text{sim}(y, \sum_{a=1}^d \gamma_\nu^a(y, [s]_a))$, and $\text{sim}(x, y) = \frac{x^\top y}{\|x\| \|y\|}$, $\gamma_\nu$ and $s$ are the same as Equation (1) with slightly change of dimension with linear transformations.

One notably difference between Equation (S9) and Equation (1) from the GCL lies in the negative sample size. In energy-based GCL, the model can treat all the examples within a minibatch as negative examples, while the original GCL contrasts with only one negative example in the minibatch by permutating labels. This refinement can benefit the learning efficiency of the NICA step, the correlation between labels decrease faster than the original GCL, as illustrated in Figure S1.

## A.5. Invertible neural network

The recent interest in generative modeling has popularized the use of *invertible neural networks* (INN) in machine learning, with prominent examples such as normalizing flows [7] and neural ordinary differential equations (ODEs) [2]. Unlike standard neural networks, an INN seeks to establish a one-to-one mapping between the input and output domains, *i.e.*, the forward map $s = f_\psi(z)$ as well as the corresponding inverse map $z = f_\psi^{-1}(s)$. Standard constructions of INN achieve representational flexibility by stacking simple invertible transformations. In practice, the efficiency of the forward or inverse passes are often trade-off depending on the application needs [6]. Here we aim for fast forward computations, thus adopt the *masked auto-regressive flow* (MAF) design for our INN [6], which allows efficient parallelization of the autoregressive architecture via *causal masking*.

Figure S1. Comparison of correlation decreasing with GCL and FDV in the MNIST dataset. FDV approaches the optimality faster, which indicates better efficiency.

Let $\{z^t\}_{t=0}^T$ be a flow of length $T$, in which $z^0 = z$, $z^{t+1} = F_t(z^t)$, $s = z^T$, and we let $f_\psi = F_1 \circ F_2 \cdots \circ F_{T-1}$. Specifically, the MAF is constructed as a series of *shift and scale* transformations $z^{t+1} = F_t(z^t) \triangleq a_t(z^t) \odot z^t + b_t(z^t)$, where $\odot$ is the element-wise product, and $a_t(\cdot)$ and $b_t(\cdot)$ are vector transformations known as scale and shift, respectively, that follow a causal autoregressive structure, *i.e.*, that $[z^{t+1}]_k$ only depends on $[z^t]_{<k}$. As a direct consequence of this structure, the MAF-based INN results in a tractable Jacobian $J(z) \triangleq |\det(\nabla_z f_\psi)|$ of $f_\psi(z)$ that facilitates likelihood computations. In fact, assuming the sources $S$ have prior density $p(s)$, the likelihood of the features $p(z)$ of the MAF specification is given by [6]:

$$\log p(z) = \log|\det(\nabla_z f_\psi)| + \log p(s) = \sum_t \log|a_t(z^t)| + \log p(s) \triangleq \mathcal{L}_{\text{FLOW}}(f_\psi) \tag{S10}$$

Jointly with $\mathcal{L}_{\text{GCL}}(f_\psi, r_\nu)$ (Equation (1) in the main text), $\mathcal{L}_{\text{FLOW}}(f_\psi)$ can be used to optimize the parameters of the de-mixing function $f_\psi(z)$.

## B. Regression for continuous labels

We can further extend the applicability of the proposed ECRT to the case of regressing continuous outcomes. While in principle, the procedures described in Sec 3 can be readily applied, we advocate coarse graining wrt label $y$ similar to what has been practiced in *sliced inverse regression* [5], especially when the feature dimension is high relative to the sample size. Specifically, we partition $y$ into different bins, and use feed the bin label as the conditioning variable in the GCL step. We still use the regression loss for the training of encoder and predictors.

## C. Implementation of Augmentation

Let $\hat{s}_i^k, i = 1, \cdots, n_k$ be the estimated source representation for the $k$-th class.

- Non-parametric augmentation: shuffling indices as described in the main text.

- Parametric augmentation: estimate $\hat{\mu}_k = \text{mean}(\hat{s}^k), \hat{\sigma}_k = \text{std}(\hat{s}^k)$, then $s^{k,aug} \sim \mathcal{N}(\hat{\mu}_k, \hat{\sigma}_k^2)$.

- Oracle augmentation: nonparametric augmentation with an abundance of class-conditional source space samples .

In Figure 9 from main text we compare the efficiency of parametric and nonparametric augmentation schemes under different minority sample size. In particular, we compute the MMD distance $\|\hat{\mu}_{aug} - \hat{\mu}_{ref}\|_\kappa$, where $\hat{\mu}_{aug} = \sum_i \kappa(\tilde{s}_i, \cdot)$ and $\hat{\mu}_{ref} = \sum_i \kappa(s_i, \cdot)$. Here $\kappa(\cdot, \cdot)$ is the Gaussian rbf kernel $\kappa(x, y) = \exp(\|x - y\|^2/2\sigma^2)$, $\|f\|_\kappa = \sqrt{\langle f, f \rangle}_\kappa$ is the RKHS norm and $\tilde{s}_i, s_i$ respectively denote augmented class-conditional samples (from few minority samples) and empirical distribution of class-conditional samples (where we use all samples from the same class that we holdout). In this example we use $2k$

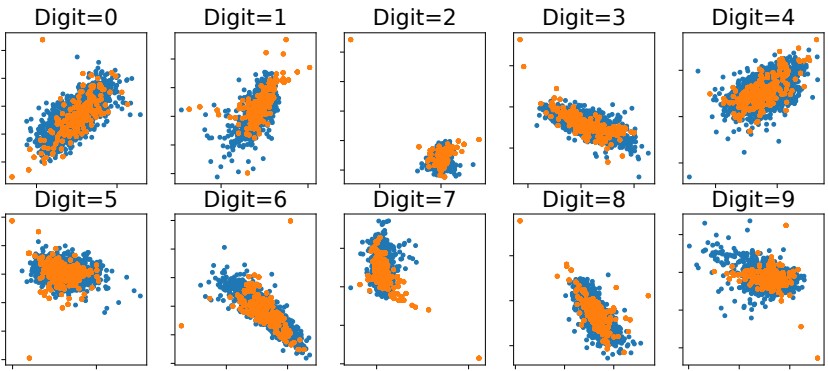

Figure S2. Feature space augmentation for MNIST.

samples for the ground truth and augment minority to the same size. Different kernel bandwidth $\sigma$ of $\kappa$ yields qualitatively similar results, and in the paper we report the one with $\sigma = 0.5$.

In Figure S2, we visualize the augmentation in feature space for the MNIST dataset. And we see for boundary points the discrepancy can be amplified by the neural network inversion, which partly explained the sub-optimal performance from feature space augmentation. In contrast, the source space augmentation advocated in this paper is more computationally efficient and robust.

Note that while in principle the majority classes can be similarly augmented, we choose not to refine our model with the augmented majorities. This decision is justified by the classical consideration for bias-variance trade-off: estimation errors of $f(z)$ is inevitable (*e.g.*, finite sample size, SGD, limited network capacity, *etc.*), and they will carry over to the augmented samples, resulting biases in the augmented estimation of our predictor. On the other hand, using augmented samples helps bring down estimation variance. For minority labels, the reduction in variance is greater than the induced bias, and consequently merits the application of ECRT to improve performance. For majority labels, this might not be the case.

## D. Toy Model Experiment

### D.1. Toy data demo

We sample seven groups of two-dimensional uncorrelated Gaussian of each with size 2000, with different means and variances as our real source representation $s$. Specifically, $s_i \sim N(\boldsymbol{\mu}_i, \boldsymbol{\Sigma}_i), i = 0, \cdots, 6$, where $\boldsymbol{\mu}_0 = [-0.5, -1], \boldsymbol{\mu}_1 = [2, 1], \boldsymbol{\mu}_2 = [5, 2], \boldsymbol{\mu}_3 = [1, 3], \boldsymbol{\mu}_4 = [-2, 1], \boldsymbol{\mu}_5 = [-3.5, 4], \boldsymbol{\mu}_6 = [-4, -1], \boldsymbol{\Sigma}_0 = [0.5, 0.5], \boldsymbol{\Sigma}_1 = [3, 1], \boldsymbol{\Sigma}_2 = [1, 2], \boldsymbol{\Sigma}_3 = [0.3, 2], \boldsymbol{\Sigma}_4 = [1, 0.2], \boldsymbol{\Sigma}_5 = [1, 1], \boldsymbol{\Sigma}_6 = [2, 0.3]$. Then we perform classical Hénon transformation $z_{(1)} = 1 - 1.4\,\tilde{s}_{(1)}^2 + \tilde{s}_{(2)}, z_{(2)} = 0.3, \tilde{s}_{(1)}$ to generate the data in feature space.

### D.2. Extreme-classification Toy Data

We sample 1000 groups of two-dimensional uncorrelated Gaussian with mean ranges uniformly sampled from range $(-4, 4)$ and standard deviation fixed to $0.1$. Validation dataset is fixed with 20 samples per-class, and the sampler per class in training dataset varies with $5, 10, 20, 50, 75, 100, 150$. The summary for extreme-classification is presented in Table S1.

Note that when the total number of categories is 10 with 20 samples per class, ERM has top-1 accuracy as high as 0.914, and the performance drops when the number of categories increasing. With 500 categories, the accuracy decreases to 0.005, and in the scenario where we presented in the main text with 1000 categories, ERM performs no better than random guessing.

Table S1. Validation Results for Extreme classification

| METRIC | NLL | | | TOP 1 | | | TOP5 | | |
|---|---|---|---|---|---|---|---|---|---|
| SAMPLE-SIZE | ERM | ECRT W/O AUG | ECRT | ERM | ECRT W/O AUG | ECRT | ERM | ECRT W/O AUG | ECRT |
| 5 | 6.9246 | 4.3884 | **3.7098** | 0.0015 | 0.0450 | **0.0760** | 0.0053 | 0.2275 | **0.3220** |
| 10 | 6.9242 | 4.0536 | **3.4752** | 0.0018 | 0.0638 | **0.0856** | 0.0060 | 0.2666 | **0.3549** |
| 20 | 6.9185 | 3.5405 | **3.3657** | 0.0016 | 0.0741 | **0.0877** | 0.0070 | 0.3126 | **0.3675** |
| 50 | 6.9240 | 3.3794 | **3.2957** | 0.0010 | 0.0834 | **0.0934** | 0.0045 | 0.3493 | **0.3835** |
| 75 | 6.9190 | 3.3499 | **3.2594** | 0.0020 | 0.0846 | **0.0939** | 0.0060 | 0.3571 | **0.3950** |
| 100 | 6.9182 | 3.3394 | **3.2675** | 0.0020 | 0.0834 | **0.0934** | 0.0053 | 0.3580 | **0.3918** |
| 150 | 6.9174 | 3.2731 | **3.2597** | 0.0011 | 0.0906 | **0.0984** | 0.0059 | 0.3777 | **0.4034** |

# E. Real-world Data Experiments

## E.1. Image Data Benchmarks

We summarized the image datasets in Table S2 and the network architectures used for respective datasets in Tables S3, S4, S5 and S6. The hyperparameters we used in these experiments are presented in S10. The results reported here are from our regularized non-parametric ECRT implementation, parametric ECRT implementation show a similar trend, with slightly improved performance (results now shown). We use 2 latent dims for MNIST and 32 latent dims for CIFAR100, iNaturalist and Tiny-Imagenet.

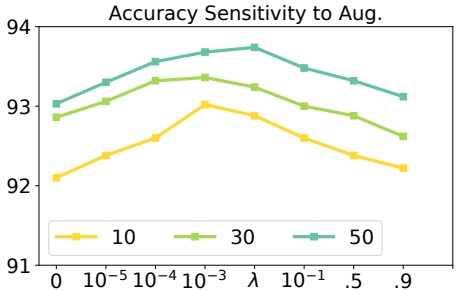

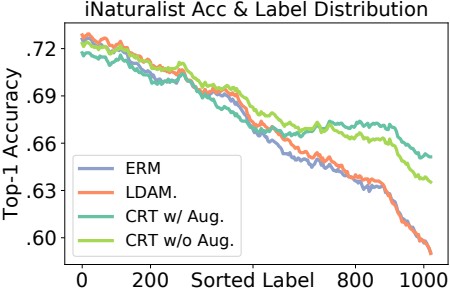

Figure S3. Sensitivity analysis (overall top 1 accuracy) of augmentation strength $\lambda$. Complementing Figure 11 in main text.

Figure S4. Class-conditional Top-1 accuracy curve for iNat2019. Complementing Figure 1 (F1 score). Note that ERM and LDAM show better accuracy for sample-rich majorities, but worse F1 scores. This evidences majority bias, that a predictor has a low specificity for data rich classes.

**Preprocessing** For iNaturalist dataset, we used pretrained Inception V3 to extract the features with dim=$2048$. For Tiny-imagenet we finetuned the Resnet 18 to extract the features with dim = $512$.

**Baselines** We used the ERM, LDAM [1], Focal [2], IW [3], and VAT [4] baseline implementations. For GAN we adopted the CGAN model with architecture shown in S9 and noise dimension listed in S11 for each dataset. We compared all the models with their own best performance after early stopping.

**Discrepancy of baseline performance.** We noticed that our implementation of baseline models, especially for the ERM baseline, yields results better than what's reported in literature (LDAM in particular). Specifically, our results look better. After carefully compared our implementation to the LDAM codebase, we see that the discrepancy comes from the choice of optimizer. The use of vanilla SGD optimizer, as practiced in LDAM, results in degraded performance of baseline, and consequently a larger performance gap compared to strong solutions.

**Majority bias.** In Figure S4, we give the top-1 accuracy wrt different minority size on the iNaturalist dataset. This figure is complementary to the F1-label frequency plot given in Figure 14 from the main text. While the improvement at the tail part are strong under both metrics, we see clear evidence of majority bias in the Top-1 accuracy plot. ERM and LDAM show better performance in accuracy for the sample abundant majority regime, but severe performance drop in the sample deficient minority regime. This is because ERM and alike finds it more rewarding to favor the majorities during inference, which gives better sensitivity but much worse specificity for the majority samples, and consistently hurting the performance for minorities.

## E.2. Language Data Benchmark

**Dataset and preprocessing.** In this experiment, have used the `arXiv` dataset hosted on `Kaggle` [5]. We use the pretrained BERT model from the `transformers` package [6] to extract sentence features. Specifically, we applied the `SciBERT` model (`allenai/scibert_scivocab_uncased`) [1], and used the BERT default 768-dimensional sentence embedding for each abstract. The training set includes 160k data, where class labels with more than 5k samples are identified as majority

---

[1] https://github.com/kaidic/LDAM-DRW

[2] https://github.com/artemmavrin/focal-loss

[3] https://github.com/idiap/importance-sampling

[4] https://github.com/lyakaap/VAT-pytorch

[5] https://www.kaggle.com/Cornell-University/arxiv

[6] https://github.com/huggingface/transformers

classes, with the rest assigned to minority label classes. All label classes with less than 20 samples have been excluded from our analysis. This gives us a total of 14 majority classes and 138 minority classes.

**Setup.** Different from the image benchmarks, the `arXiv` data is a multi-label prediction task. Each abstract is associated with at least one, possibly multiple labels, and we make binary classifications for each label class. In the training of GCL model, we allow samples with multiple labels to be reused by different classes, as each constructs a valid source IC distribution under our hypothesis. Only standard ECRT is considered in this experiment. We set the source space dimension to $64$ and use the network architecture described in Table S7.

**Evaluation.** The accuracy reported for this experiment is defined as follows: say a sample is associated with $k$-labels, then we compared the top-$k$ predicted labels to the true labels, and report the averaged accuracy for this sample. Like previous experiments, we target a balanced evaluation set. However, getting a perfectly balanced evaluation set is impossible here, as samples are associated with multiple labels. We extracted a nearly-balanced evaluation set including 847 samples, where each label has 10 to 50 counts. Most of classes have 10-15 samples in our nearly-balanced evaluation set.

Table S2. Summary of datasets

| NAME | DIM | TRAIN (MAJORITY) | TRAIN (MINORITY) | VALIDATION |
|---|---|---|---|---|
| MNIST | $(28 \times 28)$ | $6000 \times (1 \text{ OR } 5)$ (CLS) | $1200 \times (1 \text{ OR } 5)$ (CLS) | $1000 \times 10$ (CLS) |
| CIFAR | $(32 \times 32 \times 3)$ | $500 \times 50$ (CLS) | $500 \times 50$ (CLS) | $100 \times 100$ (CLS) |
| INAT | $(\text{NONE} \times \text{NONE} \times 3)$ | $(\geq 120) \times 725$ (CLS) | $(< 120) \times 285$ (CLS) | $3 \times 1010$ (CLS) |
| TINY | $(64 \times 64 \times 3)$ | $450 \times 100$ (CLS) | $45 \times 100$ (CLS) | $50 \times 200$ (CLS) |
| ARXIV | $(None)$ | $(> 5000) \times 14$ (CLS) | $(< 5000) \times 138$ (CLS) | $12 \times 152$ (CLS) |

Table S3. MNIST experiment network architecture.

| NETWORK | ARCHITECTURE |
| --- | --- |
| ENCODER | FC(UNIT=32)+ReLU
+ FC(UNIT=32)+ReLU
+ FC(UNIT=2) |
| DECODER | FC(UNIT=32)+ReLU
+ FC(UNIT=32)+ReLU
+ FC(UNIT=10) |
| FLOW | MAF($n_{blocks} = 4$,
$hidden_{size} = 128$,
$n_{hidden} = 2$). |

Table S4. Cifar100 experiment network architecture.

| NETWORK | ARCHITECTURE |
| --- | --- |
| ENCODER | RESNET18 [a]
+ FC(UNIT=32) |
| DECODER | FC(UNIT=256)+ReLU
FC(UNIT=100) |
| FLOW | MAF($n_{blocks} = 4$,
$hidden_{size} = 128$,
$n_{hidden} = 2$). |

[a]Resnet 18 without last layer

TABLE S5. INATURALIST EXPERIMENT NETWORK ARCHITECTURE.

| NETWORK | ARCHITECTURE |
| --- | --- |
| PRETRAIN | INCEPTION(V3) |
| ENCODER | FC(UNIT=1024)+ReLU
+DROPOUT(0.1)
+ FC(UNIT=512)+ReLU
+DROPOUT(0.1)
+ FC(UNIT=32) |
| DECODER | FC(UNIT=32)+ReLU
+DROPOUT(0.1)
+ FC(UNIT=512)+ReLU
+DROPOUT(0.1)
+ FC(UNIT=1010) |
| FLOW | MAF($n_{blocks} = 4$,
$hidden_{size} = 128$,
$n_{hidden} = 2$). |

TABLE S6. TINY IMAGENET EXPERIMENT NETWORK ARCHITECTURE.

| NETWORK | ARCHITECTURE |
| --- | --- |
| PRETRAIN | RESNET18 |
| ENCODER | FC(UNIT=1024)+ReLU
+DROPOUT(0.1)
+ FC(UNIT=512)+ReLU
+DROPOUT(0.1)
+ FC(UNIT=32) |
| DECODER | FC(UNIT=512)+ReLU
+DROPOUT(0.1)
+ FC(UNIT=200) |
| FLOW | MAF($n_{blocks} = 4$,
$hidden_{size} = 128$,
$n_{hidden} = 2$). |

TABLE S7. ARXIV EXPERIMENT NETWORK ARCHITECTURE.

| NETWORK | ARCHITECTURE |
| --- | --- |
| PRETRAIN | BERT |
| ENCODER | FC(UNIT=1024)+ReLU
+DROPOUT(0.1)
+ FC(UNIT=512)+ReLU
+DROPOUT(0.1)
+ FC(UNIT=64) |
| DECODER | FC(UNIT=64)+ReLU
+DROPOUT(0.1)
+ FC(UNIT=512)+ReLU
+DROPOUT(0.1)
+ FC(UNIT=152) |
| FLOW | MAF($n_{blocks} = 4$,
$hidden_{size} = 128$,
$n_{hidden} = 2$). |

Table S8. MNIST results with different numbers of minority categories

| # MINORITY LABEL | 1 | | 5 | |
| --- | --- | --- | --- | --- |
| | NLL | TOP 1 | NLL | TOP 1 |
| ERM | 0.342 | 0.933 | 0.390 | 0.904 |
| LDAM | 1.6737 | 0.9609 | 1.50 | 0.940 |
| ECRT | **0.186** | **0.972** | **0.257** | **0.950** |

TABLE S9. GAN NETWORK ARCHITECTURE.

| NETWORK | ARCHITECTURE |
|---|---|
| GENERATOR | FC(UNIT=256) +LEAKYRELU
+ FC(UNIT=256)+LEAKYRELU
+FC(UNIT=LATENT$_{dim}$) |
| DISCRIMINATOR | FC(UNIT=256) +LEAKYRELU
+DROPOUT(0.1)
+ FC(UNIT=256)+LEAKYRELU
+DROPOUT(0.1)
+FC(UNIT=1)+SIGMOID |

TABLE S10. HYPERPARAMETER OF DATASETS (ECRT)

| NAME | REG WEIGHT | AUG STRENGTH |
|---|---|---|
| MNIST | 1E-2 | 1E-3 |
| CIFAR | 1E-2 | 1E-3 |
| INAT | 5E-3 | 1E-3 |
| TINY | 1E-3 | 1E-3 |
| ARXIV | 1E-2 | 1E-2 |

TABLE S11. HYPERPARAMETER OF DATASETS (GAN)

| NAME | NOISE DIM |
|---|---|
| MNIST | 32 |
| CIFAR | 64 |
| INAT | 128 |
| TINY | 128 |
| ARXIV | 64 |