# OpenReview forum: "Supercharging Imbalanced Data Learning With Energy-based Contrastive Representation Transfer"
_NeurIPS.cc/2021/Conference — NeurIPS 2021 Spotlight_

### Official Review · Reviewer_yZHA · 2021-07-13

**Rating:** 6
**Confidence:** 4

**Summary:**

- This paper tackles the class imbalance problem in the perspective of knowledge transfer of the dominant classes to minority classes via data augmentation on the feature space. The proposed method first learns a feature extractor using the dominant classes. Based on these features, generalized contrastive learning is conducted with the masked auto-regressive flow to map the features into source space that the independency of each dimension is guaranteed with an invertible transformation. Then, a causal data augmentation procedure on the source space is conducted to enlarge the representation of minority classes. The empirical effectiveness of the proposed Energy-based Causal Representation Transfer (ECRT) is demonstrated on widely used imbalanced benchmarks with the comparison of strong baselines.

**Ethical Concerns:**

There are no ethical issues.

**Limitations And Societal Impact:**

The authors adequately addressed the limitations and potential negative societal impact of their work.

**Main Review:**

**Pros.**

- **Well motivated problem and solution.** As the many real-world datasets have imbalanced class distributions, improving DNNs under class imbalance is an important and well-motivated problem. Also, data augmentation of minority classes is definitely one of the most natural solutions, and leveraging the knowledge of majority classes to conduct it has been known to be effective in the related literature.

**Cons.**

- **Vague source of empirical improvement and insufficient ablation study.** Figure 1 shows that there is no significant difference between empirical performances of ECRT with and without the proposed augmentation. Namely, the improvement does not come from the augmentation for minority classes itself but might come from the components which are not directly related to class imbalance, such as energy-based GCL and modeling the additional classifier in the source space themselves.
Also, the presented ablation study is not enough now. The independent gain from the following components should be additionally presented in the ablation study:(1) Energy-based GCL compared to the original GCL (currently, Figure S1 implicitly reveal the improvement, but a more direct comparison with test accuracy is required), (2) Additional classifier module \phi for predicting labels from the source, (3) Restriction of feature encoder learning to the majority classes compared to the learning with all samples including minority classes, and (4) Parametric augmentation itself by comparing with non-parametric augmentation under the use of $\tilde{\mathcal{L}}_{\text{GCL}}$
- **Omitted details for reproducibility.** Many details of the proposed ECRT are currently omitted as enumerated follow; hence it is now hard to understand and reproduce the proposed method fully:
    - According to lines 155-156, the feature encoder is only trained with the majority classes ($M >2$, i.e., $m ≤ 2$). Is this selection commonly applied to all datasets? As the many real-world imbalanced datasets have a long-tailed distribution (i.e., continuously decreased number of samples), such a hard threshold seems to be sub-optimal. Also, after the pre-training phase (step 1 in Algorithm 1), is the feature encoder never trained again? Algorithm 1 implies no fine-tuning of the encoder, but it is unnatural because the feature encoder is never trained with minority classes.
    - In the proposed ECRT, there are two classifiers 1) $h_{\phi^{'}}(\boldsymbol{z})$ on features $\boldsymbol{z}$ and 2) $h_{\phi}(\boldsymbol{s})$ sources $\boldsymbol{s}$. Then, how is the prediction conducted? Just taking the average of the outputs of two classifiers? There is no description about it now.
    - No mention about sensitivity and selected value for $\rho$ in Equation (5).
    - Omitted source of pre-training for image classification in lines 362.
    - No descriptions about 1) toy model in lines 369-377 and 2) extreme classification in lines 386-390.
- **Limited methodological and empirical contributions baselines for class imbalanced learning.** The methodological contributions of this paper can be summarized into two components: (1) modification of GCL to energy-based GCL and (2) adding regularization of data likelihood with FLOW. However, the improvement from (1) is not clearly validated, and the sensitivity of hyper-parameter $\rho$ in Equation (5) is also not provided as previously mentioned. Additionally, the idea of casual representation transfer and augmenting minority samples are not the novel ideas of this paper but the ideas from the original papers. Also, the stronger baselines for handling class imbalance should be compared [1, 2], and more related works should also be added to related works [3, 4, 5].

**Other comments.**

- **Evaluation metrics.** Negative log-likelihood (NLL) is now used as one of the metrics, but its effectiveness is quite questionable. What is the behind rationale of such a choice, and how about considering other metrics for class imbalance such as Geometric Means (GM)?
- **Visualization of class distribution.** Although the authors present the descriptions about the used datasets in lines 353-360, it would be better to visualize their class distributions to improve the readability.
- **Inconsistency**
    - Both CRT and ECRT terms are now used to indicate the proposed method
    - In Table 1, there are no results about F1 score. Conversely, Figure 9 and 11 only report F1 score without any motivation of such choice. It would be better to unify the used metric, e.g., additional results of F1 score to Table 1.

[1] Kang et al., Decoupling Representation and Classifier for Long-Tailed Recognition, ICLR 20 \\\
[2] Menon et al., Long-tail learning via logit adjustment., ICLR 21 \\\
[3] Tang et al., Long-Tailed Classification by Keeping the Good and Removing the Bad Momentum Causal Effect., NeurIPS 20 \\\
[4] Hariharan and Girshick., Low-shot Visual Recognition by Shrinking and Hallucinating Features., ICCV 17 \\\
[5] Kim et al., M2m: Imbalanced Classification via Major-to-minor Translation., CVPR 20

**Time Spent Reviewing:**

6 hours

---

> ### Author Response · Authors · 2021-08-10
> **Clarifications on encoder, pre-training, significance and evaluation**
>
> We thank the reviewer for reviewing the paper and provide valuable feedback, please see our point-to-point responses below. Replies to shared comments can be found in our general responses.
>
> *Q: Is the feature encoder training only apply to the majority classes?*
> * For the toy experiments, yes. For the large-scale, real-world experiments, we have used off-the-shelf general-purpose pre-trained encoders (i.e., ResNet and BERT), which follows common practice in the literature. Note that these encoders do not necessarily are trained in a supervised fashion (e.g., can be replaced with SimCLR trained encoders). These pre-trained encoders are more reliable and should be favored whenever possible.
>
> *Q: After pre-training, is the feature encoder never fine-tuned again?*
> * Yes, the base-feature encoder is fixed after pretraining. This is because if we fine-tune the base-encoder, the base-feature distribution p(z) will change. Consequently, the source-feature S derived based on the original feature distribution will no longer be conditionally independent, which compromises the validity of the subsequent causal augmentation step. While an end-2-end solution that jointly trains base-encoder and GCL is feasible, we do not recommend it. Because we have found the GCL adapts slowly to the changes of base-encoder, and it does not perform as well as the two-stage training without extensive manual tuning. As an alternative, we can consider alternating the base-encoder training/fine-tuning with GCL training. However, we are not enticed to do so because: (a) the result without fine-tuning the base-encoder already consistently leads the performance chart; (b) improvements are marginal; (c) GCL adapts slowly to the new base feature distribution and the cost does not justify the gains.
>
> *Q: In the paper, there are two classifiers: feature predictor h(z) and source predictor h(s). Then how is the prediction conducted?*
> * This is major confusion about ECRT, which we are happy to clarify. Note that the feature predictor h(z) is only used for the encoder-pretraining, and all final predictions are made using the source predictor h(s). This is because the training of source predictors enjoys all the benefits from causal-representation engineering, such as feature pre-whitening and causal augmentation (see Sec 3.4 for details). These are not available to the original feature predictor h(z). In fact, modeling with (causally-disentangled) source representation S instead of the original (supervised-learning based) feature representation Z is a view we want to promote through this work.
>
> *Q: Significance of this work*
> * We want to emphasize that the development of the energy-based GCL algorithm is significant on its own, extending well beyond the scope of imbalanced data learning. Note GCL is a key algorithm for non-linear independent component analysis that is core to many other applications, so any improvements may imply far-reaching impact. The connection we made to mutual information estimation is non-trivial as well. Since improving GCL is not the main problem of this paper, we defer the GCL performance comparison to the Appendix (Fig. S1). We will add the accuracy comparison in our revision, along with more ICA metrics and wall time comparisons.
>
> * Note that while causal augmentation is first proposed in [Teshima, et al. (2019)], the original solution only applies to few-shot learning and it does not work well for complex data distributions such as image and text. The work presented here has described the necessary modifications needed to enable efficient learning with complex data.
>
> *Q: Comparisons and discussion of the more recent strong baselines suggested [1,2,3,4,5]*
> * Thanks for suggesting these strong baselines. We have carefully read the papers suggested, and we are unable to directly compare the performance to these solutions given the limited time rebuttal phase allows. This is the experiment setups are very different from what we have adopted here (different datasets and architectures), and extensive work is required to make sure the modeling choices are comparable to ensure a fair comparison. For example, [1] is an exhaustive comparison of different combinations of solution strategies. And as noted in our general response, the consistency perspective promoted by [2] may not be appropriate for imbalanced learning. Nevertheless, we are still trying our best to accommodate the reviewer's suggestion, and hopefully, we can have some results soon. Our work is in a similar flavor to [3] which also explores reasonable causal assumptions. We also think some ideas in [4,5] are interesting and will add to our discussions.
>
> *Q: Missing visualization of class distributions for real-world dataset*
> * Actually we provide visualization of the class distribution of the iNaturalist dataset in Fig. 1 (shaded region, referred to as label frequency in the figure). This is a very representative case of real-world dataset. We can accommodate the reviewer's request by adding the class distributions for all the datasets we used in the Appendix due to the space limits on the main text. In Fig. 8 we have the feature distribution extracted by our algorithm visualized.
>
> *Q: Evaluation metric*
> * Thanks for the suggestion. We note this comment is related to the question on calibration made by 8aQQ, which also suggests alternative metrics. Our detailed reply can be found in the general response. Actually, since we have adopted the imbalanced-training & balanced-evaluation setup for our experiments, the results are not very sensitive to the metrics such as NLL -- as they have already been properly balanced. This is also the goal sought after by the geometric-mean suggested by the reviewer because GM effectively rebalances the importance of sensitivity between majorities and minorities. We argue the class-F1 score provided in Fig. 1 is in fact much more informative than the bulk statistics such as accuracy, NLL and GM: it not only accounts for both sensitivity and specificity but is also tailored to each individual class. We encourage the reviewer to focus on this metric.
>
> *Q: notation consistency*
> * Thanks for pointing out these minor issues. We have fixed the notation consistency and added the F-1 results to the Table. The motivation for using F-1 is given in our response to the question on the evaluation metric. Note that we do not report F-1 in the Table following the practice of LDAM and FOCAL paper, where the focuses are given to accuracies in their Tables.
>
> *Q: No descriptions about 1) toy model in lines 369-377 and 2) extreme classification in lines 386-390.*
> * Detailed descriptions for the toy models we used have been given in line 349-352 and Appendix Sec D.

---

> > ### Comment · Reviewer_yZHA · 2021-08-26
> > **Remaining Concern about Question 1**
> >
> > Thank you very much for the response. I appreciate the effort that the authors put into addressing my questions. I believe that the above results and discussion can significantly improve the quality of the manuscript. However, the major concern about **"Vague source of empirical improvement and insufficient ablation study"** is not addressed yet. Therefore, I will keep my initial score.

---

> > > ### Author Response · Authors · 2021-08-26
> > > **Oops! We did not intentionally leave out this major concern...**
> > >
> > > Dear reviewer,
> > >
> > > Thank you for your response! We want to apologize for our negligence, response to that specific comment was inadvertently lost while we compile the overall rebuttal. I do not currently have access to the computer that stores the original reply (and the numbers for that), so I will just give a very quick response to the points that you find unsatisfactory.
> > >
> > >
> > > *Q: Fig. 1 shows that there is no significant difference between empirical performances of ECRT with and without the proposed augmentation*
> > > * We respectfully disagree with this evaluation. The difference is very pronounced in the region where minority samples are scare. Also, we refer the reviewer to Fig. 9 and Fig. 11 on the abaltion study on the effectiveness of augmentation. Fig. 6 also shows augmentation is drastically improved performance for the extreme classification (again, epscially in the low-sample regime). Fig. S3 and S4 also contains ablation for augmentation. We are very thorough on how augmentation improves performance.
> > >
> > > *Q: suggested ablations*
> > > 1. Energy-based GCL versus the original GCL. The primary gain of our energy-based formulation is the learning stability & efficiency, which is partly covered in Fig. S1. When carefully tuned, the original GCL has a similar performance in terms of accuracy, but energy GCL learns faster and more stably.
> > > 2. We believe this is a misunderstanding of our architecture as detailed in our reply to your comment "Then how is the prediction conducted?" above. Basically, ERM is the predictor using the original feature $z$, and ECRT w/o and w/ augmentation are the ones based on the source predictor $h_{\phi}(s)$. So this ablation is essentially in the resported results.
> > > 3. While not exactly what the reviewer asked for, the reviewer can for now refer to Table S8, where the encoder is trained on different numbers of majority classes. The important message here is that proper causal assumption strengthens the feature encoders trained only on the majority data, which is what makes robust generalization possible. While we will come back with numbers later, we caution, how including minority examples in the prre-training of feature encoder differs case-by-case. They may not affect performance at all (majority dominace), improve performance (predictive features consistent with those used by majority) and completely devastating (containing spurious features, overfit). We do not want to suppement a number that might mislead the readers, that is why we decide to keep what are reported in our original manuscript, but leave out the ablation for training with all examples. Also, we want to stress that all our large-scale real-world experiments are based on general-purpose pre-trained feature encoders (e.g., using self-supervised learning rather than supervised learning) have shown significant gains for ECRT over competing baselines. This is a more realistic application scenario that we want to focus on. That said, we will give some ablations that the reviewer have requested.
> > > 4. This ablation is actually reported in Table 1, ECRT is the original non-parametric augmentation while ECRT-MULTI is the parrametric augmentation (since different Gaussian priors are used for each class).

---

> > > > ### Comment · Reviewer_yZHA · 2021-08-29
> > > > **Few More Questions**
> > > >
> > > > Thank you for the detailed response. Most of my concerns are now addressed, except few remaining vague points.
> > > >
> > > > 1. Although the authors highlight the improvement from their method (Fig 1, 9, 11) based on their own metric (class-wise F1 score),  I wonder about the improvement based on the classical metric, i.e., class-wise accuracy. Can the authors provide such results? As many related works usually report the performance based on the classical metric, these results would be necessary for the comparison.
> > > >
> > > > 2. Also, the author said that "...general-purpose pre-trained feature encoders (e.g., using self-sup learning)...". Where is the detailed description of the general-purpose pre-training method especially in large-scale real-world experiments?

---

> > > > > ### Author Response · Authors · 2021-08-30
> > > > > **We have the numbers and other info you asked for!**
> > > > >
> > > > > Dear Reviewer yZHA,
> > > > >
> > > > > Thank you for getting back to us! We are really sorry that after retrieving the original rebuttal draft we prepared, we realized that some of the replies to you have been inadvertently dropped due to careless copy-pasting (including the ablation, F1, and pertaining details.) Please accept our sincerest apology, we should be more careful proofreading our replies. Thanks for this new reviewer-author discussion round so that we still have a chance to clarify. Please find our answers below.
> > > > >
> > > > >
> > > > > *Q: Class-wise F1 score is highlighted and it looks good, how about the classical metrics such as class-wise accuracy and F1 score.*
> > > > > * For comparison of the class-wise accuracy, please see Fig. S4. Note that we have argued class-wise is not a proper metric here, although widely employed in the literature. This is because the class-wise accuracy will be biased for the majority classes (e.g., ERM has superior accuracy for the majority classes but terrible accuracy for minority classes relative to imbalanced learning schemes, which makes it difficult to interpret the results).
> > > > >
> > > > > * We note that neither LDAM nor FOCAL has reported the F1 score in their papers (they only focused on accuracy). Nevertheless, we computed the overall F1 scores for these models. For the Cifar100 dataset, we have the following results (using the model reported in Table 1):
> > > > > * ERM: F1=0.439
> > > > > * SMOTE: F1=0.444
> > > > > * FOCAL: F1=0.391
> > > > > * LDAM: F1=0.408
> > > > > * ECRT: F1=0.482
> > > > >
> > > > > FOCAL and LDAM gave very poor F1 scores in this case, even worse than the ERM baseline. We have doubled checked our implementation and made sure we have done it right. SMOTE improved ERM, but the largest gain is obtained by our ECRT.
> > > > >
> > > > > *Q: Where is the detailed description of the general-purpose pre-training method especially in large-scale real-world experiments?*
> > > > > * For TinyIMAGENET, we use [this pre-trained ResNet-18](https://download.pytorch.org/models/resnet18-5c106cde.pth) from the official PyTorch model repository; for iNaturalist, we use [this pre-trained Inception-V3](https://download.pytorch.org/models/inception_v3_google-1a9a5a14.pth) from the official PyTorch model repository; For the arXiv dataset, we used [this SciBERT *allenai/scibert_scivocab_uncased*](https://github.com/allenai/scibert) from the official Hugging Face library. Clarifications of these pre-trained encoders will appear in our revision.
> > > > >
> > > > > * And we need to make a correction to the experiment setup reported in the paper: the results in Table 1 for iNaturalist is obtained using the Inception-V3, not ResNet-18. We did use ResNet in our early experiments but then switched to Inception-V3 to be consistent with some prior studies.
> > > > >
> > > > > **Following up on the last reply on ablation.** Although the reviewer finds our arguments in the last reply to the ablation convincing, here are the numbers that we have promised to fetch.
> > > > >
> > > > > *Q: How about the comparison of feature encoder trained with the majority only and both majority & minority.*
> > > > > * Table 1 reports the results for the majority only trained feature encoder. For both majority & minority trained encoder, we have Cifar100 top1=51.79, top-5=81.02, NLL=2.03. This is slightly worse than the majority-only result but still outperformed other competing solutions.
> > > > >
> > > > > *Q: Performance difference between energy-based ECRT and GCL-based CRT*
> > > > > * Between the two, top-1 & top-5 difference < 0.1, NLL difference <0.2 on the Cifar100 dataset, so the difference is not significant between the two. By comparison, that ECRT outperformed the best competing solution by 2$\sim$2.5 (ECRT & ECRT-MULTI, respectively) on Top-1 Acc and 2.5$\sim$3.5 on Top-5 Acc.

---

> > > > > > ### Comment · Reviewer_yZHA · 2021-09-01
> > > > > > **After rebuttal**
> > > > > >
> > > > > > Thank you again for the detailed responses. My major concerns are mostly addressed; hence I raise my score to 6 from 5. However, it's quite surprising that FOCAL & LDAM show poor F1 scores even worsen than ERM. Please add discussions about these results along with the superiority of F1 score as the authors previously responded.

---

### Official Review · Reviewer_sKzi · 2021-07-16

**Rating:** 7
**Confidence:** 3

**Summary:**

The paper proposes to use an auxiliary feature space (S) via non-linear independent component anaylsis (NICA) on an existing feature space (Z), e.g., of a pre-trained deep neural network, to transfer causal information of majority classes to minority classes in the context of imbalanced (or long-tailed) classification. More specifically, the method trains an invertible neural networks Z → S using only majority samples to perform NICA, and uses this model to augment minority samples by simply permuting their S-features, thanks to the coordinate-wise independence of S. This augmentation has an effect of transferring the causal features of majority samples into the minority samples. Experimental results confirms that such techniques can indeed prevent classifiers from overly relying on non-generalizable features of minority samples.

**Limitations And Societal Impact:**

The paper does not clearly states their limitations or societal impacts in their text, so I suggest the authors to add the respective paragraphs in the revision.

**Main Review:**

Overall, I found the paper is clearly written with novel idea to tackle an important topic of imbalanced classification. I appreciate that the introduction is carefully put to articulate the messages the authors want to convey, and that the paper properly discusses the challenges and limitation of the current method, and several ideas to alleviate them. Although someone might concern that the proposed method is too complex in practice, or with increased training complexity, but I think it is OK to me in a sense that each of the components is generally well-motivated and contains many interesting idea, giving us more insights on leveraging NICA in the context of deep learning. The experimental results are also promising, compared to the state-of-the-art results of LDAM, although I slightly feel that the paper could have also covered some other recent approaches other than LDAM in their comparison to give readers a more clearer view.

**Time Spent Reviewing:**

4

---

> ### Author Response · Authors · 2021-08-10
> **Thanks for the very positive comments**
>
> We thank the reviewer for the very positive comments. You can find discussions on the potential limitations of ECRT in lines 291-297. We will add discussions as comparisons to some more recent works as suggested by the reviewer (other reviewers have provided a list of good candidates). We will further stress the societal impact of our research, which is significant due to its wide applicability.

---

### Official Review · Reviewer_8aQQ · 2021-07-16

**Rating:** 6
**Confidence:** 4

**Summary:**

This paper consider the problem of learning in the presence of a severe class imbalance, specifically when there are more than two classes. It stipulates a causal generating mechanism to "enable efficient knowledge transfer from the dominant classes to the under-represented counterparts," which hinges on the important assumption that all classes share a common set of salient features. In detail, the proposed procedure involves: (1) pre-training an encoder and predictor, (2) NICA estimation using the learned features, (3) source-space augmentation, (4) minority predictor modeling with the augmented source. Experiments on CIFAR100, iNaturalist, TinyImagenet and ArXiv show that the model

**Limitations And Societal Impact:**

Yes.

**Main Review:**

There is a lot going on in this paper, and it took a little while to wrap my head around it. One of the reasons is the organization of the paper, where the details of the proposed approach are spread over many sections. It may have been easier to understand the proposed approach if the final version was presented first, followed by further details about what didn't work as well (perhaps relegated to appendix). As is, several different versions of different aspects of the pipeline are presented in different parts of the paper, and I would like to see a consolidation the of the presentation around the best ideas.

Another issue, which is also a strength of the paper, is that this work draws on many recent ideas - normalizing flows, NICA, GCL, FDV, MI, etc. - which means that much *necessary* background introduction is relegated to passing citations. This paper is far from self-contained, and I found this particularly egregious in 3.3, which is very dense. The discussion of MAF is also confusing, where it is at times described as "efficient" (3.2), and others as "costly." That being said, I found 3.4 to be quite helpful in clarifying the contributions.

Perhaps the most critical step of the proposed approach is the pre-training of the encoder and predictor. Because all subsequent steps rely on this one (cf Assumption 3.1), if anything goes wrong at this stage (e.g. overfitting) one would expect everything else to fail. As such, I would have liked to more experimental ablations and discussion of this step. For example, it would be interesting to know how regularization at this stage affects downstream performance.

Other Concerns:

1) There is insufficient detail about how the baselines are tuned. For example, Focal loss has two important hyper-parameters: were they both tuned? In what range?

2) The notion of discriminatively pre-training a feature extractor bears some resemblance to the recently proposed Supervised Contrastive Learning paper (https://arxiv.org/abs/2004.11362), which would have been a good further baseline, particularly if combined with a Focal loss for the classifier-learning stage.

3) The results across four datasets in Table 1 are fairly convincing (with the caveat about the baseline tuning).

Questions:

1) Can you comment on calibration beyond NLL? For example, have you computed ECE? Does the proposed procedure result in a consistent probability estimator?

2) You suggest that auto-encoders might be used to learn features. Have you tried it? Generative feature representations strike me as being somewhat different from the discriminative ones used throughout the paper, and it seems dubious that that they would satisfy A3.1 or work with ICA.

3) An important class of imbalanced problems are binary, e.g. credit card fraud, cancer detection, etc.  (In fact, this may be what comes to mind for many when talking about "imbalanced data learning"!) Can you comment on how the proposed method might work there?

** Update after author response period ** Thanks to the authors for their replies. I remain positive about the work but keep my original rating.

**Time Spent Reviewing:**

3

---

> ### Author Response · Authors · 2021-08-10
> **Responses to comments on technical details, alternative pre-trained encoder, SCL, tuning, evaluation, significance and etc.**
>
> We thank the reviewer for the positive review and constructive feedback. The comments are very specific and professional, which we have carefully replied to below. For shared comments, more detailed responses can be found in our general response section.
>
> *Q: Why is MAF both "efficient" and "costly"?*
> * This is because MAF is compared to different competing solutions in the respective contexts where the terms “efficient” and “costly” appeared. MAF is costly compared to non-invertible neural networks, which are used to construct standard feature encoders that compress data representations. We have to use MAF here because the GCL algorithm depends on invertible transformations. However, compared to other competing invertible neural networks, such as the neural ODE [Chen et al. (2018)] and Invertible Residual Networks [Behrmann et al. (2018)], MAF is highly efficient. We note that nowadays invertible neural networks can be used as off-the-shelf packages so a very detailed knowledge of their implementation is not necessary to implement our solution from scratch. Moreover, we did try other more advanced (but costly) invertible nets (e.g., spline flows [Durkan, et al. (2019)]) and saw improvements in training (being able to work with larger step sizes and learn better in higher dimensions).
> * [Chen et al. (2018)] Neural Ordinary Differential Equations. NeurIPS 2018
> * [Behrmann et al. (2018)] Invertible Residual Networks. ICML 2019
> * [Durkan, et al. (2019)] Neural Spline Flows. NeurIPS 2019
>
> *Q: The strength of this paper is that it draws on many recent ideas, which makes it difficult to be self-contained.*
> * The key idea of this paper is to leverage causality to enable the extraction & augmentation of stable features to improve imbalanced learning. We recognize it is a bit hard to follow all the technical points, as many tools have only become available very recently. For completeness, we have tried our best to provide adequate coverage of technical background within the space allowed. Our work intends to promote the exposure of these advanced statistical techniques to the machine learning community, to popularize their use and inspire new research. Hopefully, with time, these techniques will become part of the standard toolkit of machine learning.
>
> * Importantly, we note that our work not only organically integrates these techniques but also developed some improved versions of them (e.g., the more efficient energy-based GCL). Given that these techniques are the fundamental building blocks for many important applications, the significance of our work extends well beyond the scope of imbalanced data learning. We will highlight this point in our revision.
>
> *Q: Can you comment on calibration beyond NLL? For example, have you computed ECE? Does the proposed procedure result in a consistent probability estimator?*
> * Excellent point. To clarity, we believe calibration is a concept not well suited for the imbalanced learning setup. This is because imbalanced learning focuses more on the scenarios where the accurate classification and generalization of minority class examples are significant (e.g., for being associated with higher costs, see our Introduction). Also, the frequencies of the minority categories are often not well defined: either due to the observation/selection bias in the training data (e.g., users can selectively report fraudulent activities: those with minor financial losses are often not reported), or the potential data shift upon model deployment (e.g., frequencies dynamically dependent on external environments, such as seasonal changes). Those are exactly the reasons why most imbalanced learning studies have adopted imbalanced-training & balanced-evaluation in their experiments, which we also follow in this work. We can provide statistics like ECE as requested in our revision, but they are not particularly meaningful compared to the accuracy and NLL provided in the current presentation, which assess the relative confidence of accurately predicting the minority examples.
>
> * As for consistency, the causal augmentation procedure is known to be consistent (a proof can be found in [Teshima et al. (2019)]). Again we want to note that this concept does not apply to the imbalanced learning setting for reasons we discussed above.
>
> *Q: Have you tried generative models such as VAE for feature pre-training?*
> * Good question. We did try VAE as the feature encoder in the development of our algorithms, it also worked, but suboptimally. We choose not to use this strategy in the main experiments for good reasons: (1) unsupervised feature encoding schemes such as VAE usually encode redundant information that is needed for image reconstruction, but not useful for label prediction, which increases the computational burden for GCL setup as higher-dimensional feature space is required to produce the same level of accuracy; (2) also, in our pilot experiments, VAE based pretraining performed suboptimally; (3) additionally, for fairness, not all competing solutions can be rigged to accommodate a VAE arm.
>
> * Related to the reviewer's comment, we believe that adapting our ECRT framework for unsupervised representation learning is a very promising direction given the recent huge success of unsupervised pretraining (e.g., SimCLR, MoCO, BYOL, etc.). However, this is a challenging task and substantial changes are entailed (in assumptions and algorithms). We will potentially include some preliminary results in our revised manuscript in this direction. This will be more meaningful than reporting some results by simply switching out the supervised pretraining with VAE.
>
> *Q: Insufficient detail about how the baselines are tuned*
> * Thanks for this comment. We are thorough for baseline tuning, and will add the following details on how we tune the baselines to the updated Appendix:
>
> * Hyper-parameter tuning for baselines: All baselines are tuned for best performance for top-1 accuracy on validation. For FOCAL, we follow the suggestions made in the LDAM codebase and tune $\gamma \in [0.1, 1]$ while fixing $\alpha$ to the class frequency. For LDAM, we have varied the scale parameter $s$ in the range $[20,40]$ based on suggested value $s=30$ from the LDAM paper. We also found LDAM using Adam optimizer produces better results compared to the SGD practiced in the original paper. We conjectured the original paper uses SGD because theoretical guarantees for LDAM only hold provably under SGD. The difference between ERM and LDAM under Adam is less significant compared to the results reported in the original paper.
>
> *Q: Supervised Contrastive Learning (SCL) [Khosla, et al. (2020)] seems very relevant, can you comment and add to the baseline?*
> * Thank you for bringing this work to our attention. We have carefully read the SCL paper and note the following key difference to our work: while SCL’s training objective seems to be the same as ECRT’s GCL implementation, the specific designs used by ECRT in the network architecture and scoring function allows ECRT to provably disentangle feature representations, which is then exploited in the causal augmentation step in ECRT. In contrast, SCL is proposed based on the conjecture that supervised contrastive training will automatically cluster data representations of the same label: and we know from the theory of GCL this conjecture is only partly true. We have experimented with SCL (for both balanced and imbalanced datasets) as suggested by the reviewer, while it works reasonably well for the balanced cases, it is not competitive in the imbalanced setup (worse than the ERM baseline). We can explain this observation with the following argument: SCL essentially performs (generalized) self-supervised contrastive learning per label-class, which is expected to work well only with sufficient examples and it is not the case for minority classes. The fact SCL is only partially using label information makes it underperform ERM. On the other hand, ECRT exploits additional causal assumptions that allows the encoder to only retain features transferable from majority classes.
> * [Khosla, et al. (2020)] Supervised Contrastive Learning. NeurIPS 2020
>
> *Q: How the proposed framework can be used for imbalanced binary classifications*
> * Excellent question. The proposed ECRT can extend to the binary classification setting by slightly adjusting the assumption we made. First, we would like to note the direct application can also work, as long as the minority class has adequate samples -- simply use both classes in the GCL setup. A more sensible strategy is to artificially break the majority class into a few pseudo-classes --  reasonable choices can be based on auxiliary labels (e.g., zip code, job type, etc.) or cluster structures that are known to be meaningful. Now we can proceed with the GCL step as if there are multiple classes as assumed in the main text. After the GCL feature extraction, we train the original binary classifier along with the causal augmentation and other adjustment techniques proposed in ECRT.

---

### Official Review · Reviewer_weGp · 2021-07-20

**Rating:** 6
**Confidence:** 4

**Summary:**

The paper presents a solution for imbalanced classification problem – which a well-known, but otherwise difficult machine learning problem, especially in high-dimensional space. The difficulty typically lies in the unknown nature of minority classes. Naïve methods such as resampling would easily overfit. A plausible approach would impose certain assumptions, such as the ability to ‘disentangle’ the feature representation; and utilising recent identifiability result in nonlinear ICA this paper assumed that the feature representation of a data point can be decomposed to several independent source features conditioned by the data point’s class label. Also, these source features can be effectively extracted from data by a general contrastive learning process. The author carefully designed a framework and a learning algorithm to achieve this goal.

In short, the key contribution is a combination of exploiting recent results in CMT (casual transfer mechanism, but limited to regression), generalize contrastive learning (GCL) and strategy to improve training. This is to learn from an input x to its source representation s = f(z) where z is the usual feature encoder. Now, elements of s are conditionally independent given the label. From here, enhancing minority class is fairly standard.


**Main Review:**

Strength:
-	I like the general technical thoroughness of this paper, especially its attempts to discuss its shortcomings, relation to practical considerations. The supplementary materials have also made a reasonable effort to make the work as complete as possible.
-	The exploitation of the results from GCL in [1] and CMT [66] to address the imbalanced problem is welcoming.
-	I can’t say for sure due to my lack of hands-on experience, but the new energy-based CGL, exploiting ratio density trick via Donsker-Varadhan seems very useful, especially when dealing with large number of classes as claimed in the paper.

Weaknesses:

-	Despite empirical results reported, what missing is insight into when the proposed approach will fail? What assumption on the number of minority labels for it to work? What is the effect of correlations between label of minority classes and majority classes? (e.g., if have lots of major labels for horse, it might be easier to correct for minority class for zebra than for car)
-	It is also well-know that the complexity and dimensionality of the data space is important. Having said that, a popular baseline SMOTE (Synthetic Minority Oversampling Technique), which can be applied on the latent space z, is missing. And while it is understood that CMT originally designed for regression, is it possible to twist it as baseline (since it is closest to this work) ?
-	For the augmentation step, the new samples are created by randomly mixing of existing sample (in the naïve setting) or sample from a distribution. Although it is showed in figure 5 that these samples’ distribution as the ground truth distribution, there seems to be nothing to guarantee that the new created data point has the same label as the data points used to created it. Please clarify


**Time Spent Reviewing:**

5

---

> ### Author Response · Authors · 2021-08-10
> **Clarification of when the ECRT might fail, answers to overlap of representations and added baseline results**
>
> We thank the reviewer for the positive feedback and constructive comments. For shared comments, detailed responses can be found in our general response section. For individual comments, we have carefully replied below.
>
> *Q: What missing is insight into when the proposed approach will fail?*
> * We would like to point the reviewer to lines 291-297 for the discussion on the potential limitations of our proposed ECRT. To briefly recap and elaborate, the efficiency of ECRT only applies if predictive features are shared between minority and majority classes (i.e., arising from the same data generating mechanism). If there are features unique to the minority classes, then solutions based on alternative assumptions might be more appropriate (c.f., reference reviewed in line 44-50). Also, the proposed ECRT can not be directly applied to binary classification unless either auxiliary classes labels are presented or minority examples are abundant. It is important to note that real-world data is complex and an ensemble of models based on different assumptions usually work most robustly. We contribute this work as a valuable addition to the model pool, which performs strongly on its own for a diverse set of real-world datasets we have tested.
>
> *Q: Can you add baselines for SMOTE and CMT?*
> * This is a good point. We did test SMOTE in the preparation of this paper but excluded its results from the final results, as we wanted to highlight the comparison to more recent works. Specifically, SMOTE is valid when the class-conditional feature distribution is uniformly distributed in a convex region. For complex feature distributions, it can yield unsensible augments (i.e., linear interpolation might not lie on the data manifold). Although it did help improve over the naive ERM baseline and comparable to some of the SOTA models, it was less competitive to the ECRT model we have proposed. For instance, on Cifar100, SMOTE achieved Top-1 Acc: 51.42, Top-5 Acc: 79.05, and NLL 2.44. In comparison, the Top-1 Acc for other major models are ERM 49.29, LDAM 50.46 and ECRT 53.00 (see Table 1). Results for other datasets have a similar trend. Note that we followed the same hyper-parameter tuning for SMOTE. We will update SOMTE to our revision as suggested.
>
> * Regarding the reviewer’s request on including CMT as baseline, we note this paper is about how to generalize CMT beyond regression. In other words, our ablation study can be considered as the requested comparison to the CMT baseline, for the modifications we have added to fix the stability issues of the original CMT on complex data and the more efficient energy-based GCL algorithm. We can definitely use an early implementation of our model that applied the standard generalized contrastive learning (GCL) algorithm as the “CMT baseline”, which learns less efficiently but still outperforms competing solutions other than the final ECRT model.
>
> *Q: There seems to be nothing to guarantee that the new created data point has the same label as the data points used to create it. Please clarify*
> * Good question. The short answer is: no it does not, because class-conditional feature distribution can overlap across different classes, however that is actually a good thing (we will elaborate shortly). Informally, it is the overlap of features that enabled the identification of the data-generating mechanism, which allows the GCL algorithm to provably learn the invariant features. For any point in the feature space, it may pertain to a few different classes associated with different probabilities (which the model tries to predict). To make this concrete, consider pictures of horses and zebras. For a blurry image that only captures the rough shape of the animal, it will be encoded into a point in feature space that is associated with both horse and zebras (with different probability though). The right-most plot in Fig. 2 visualizes such scenarios.

---

> > ### Comment · Reviewer_weGp · 2021-08-29
> > **thanks for additional clarification**
> >
> > I would like to thank the authors for additional clarification from my review. In general, I remains positive, but I would like to keep my scores.

---

### Author Response · Authors · 2021-08-10
**General Response to the Reviewers**

We thank all reviewers for their time to provide this constructive feedback. We are glad all reviewers have found this work interesting. Given the intense technical background of our solution, we understand some confusion is bound to arise, which we are happy to clarify. We reply to the shared comments in the general responses below, and point-to-point responses to each individual comment can be found after each review.

**The significance of this work extends beyond imbalanced data learning.** Importantly, we wish to emphasize that our work not only organically integrates these techniques but also developed some improved versions of them (e.g., the more efficient energy-based GCL). Given that these techniques are the fundamental building blocks for many important applications, our work has the potential to deliver impacts well beyond the scope of imbalanced data learning. We will highlight this point in our revision. In particular, the development of the energy-based GCL algorithm is very significant on its own, extending well beyond the scope of imbalanced data learning. Note GCL is a key algorithm for non-linear independent component analysis that is core to many other applications, so any improvements may imply far-reaching impact. The connection we made to mutual information estimation is non-trivial as well. Since improving GCL is not the main problem of this paper, we defer the GCL performance comparison to the Appendix (Fig. S1). We will add the accuracy comparison in our revision, along with more ICA metrics and wall time comparisons.

**Comments on the performance evaluation with alternative metrics.** Naturally the reviewers ask why we have focused on accuracy and NLL in this work, and wonder if alternative concepts and metrics such as statistical calibration and geometric mean are more appropriate. To clarity, many standard machine learning concepts may not be well suited for the imbalanced learning setup. This is because imbalanced learning focuses more on the scenarios where the accurate classification and generalization of minority class examples are significant (e.g., for being associated with higher costs, see our Introduction). Also, the frequencies of the minority categories are often not well defined: either due to the observation/selection bias in the training data (e.g., users can selectively report fraudulent activities: those with minor financial losses are often under-reported), or the potential data shift upon model deployment (e.g., frequencies dynamically dependent on external environments, such as seasonal changes). Those are exactly the reasons why most imbalanced learning studies, with representative examples such as LDAM and FOCAL, have adopted imbalanced-training & balanced-evaluation in their experiments, which we also follow in this work. We can provide statistics like ECE as requested in our revision, but they are not particularly meaningful compared to the accuracy and NLL provided in the current presentation, which assess the relative confidence of accurately predicting the minority examples. Actually, since we have adopted the imbalanced-training & balanced-evaluation setup for our experiments, the results are not very sensitive to the metrics such as NLL -- as they have already been properly balanced. This is also the goal sought after by the geometric-mean suggested by the reviewer because GM effectively rebalances the importance of sensitivity between majorities and minorities. We advocate the use of class-F1 score provided in Fig. 1 is in fact much more informative than the bulk statistics such as accuracy, NLL and GM: it not only accounts for both sensitivity and specificity but is also tailored to each individual class. We encourage the reviewers to focus on that metric. That said, we will include some alternative metrics suggested by the reviewers in our revision.

---

### Author Response · Authors · 2021-08-25
**We are anxiously waiting for your further engagements, please initiate a discussion if additional clarification is needed**

To Dear Reviewers,

Greetings~ We thank all reviewers for putting their valuable time into evaluating our work. We are happy to see our work is viewed positively, and we appreciate all the constructive feedback received. While we understand everyone is busy, we have been anxiously waiting for additional inputs since the discussion phase kicked off. Per our experience, despite the best of efforts, confusion might not be adequately addressed in a single round of interactions. This year's NeurIPS offered the opportunity of discussion to further engage author(s) and reviewers, and we are happy to address any lingering confusion that the reviewers might have. The clock is ticking, and we wish to hear from you soon.

With anticipated thanks.

Author(s)

---

### Decision · Program_Chairs · 2021-09-27

**Decision:**

Accept (Spotlight)

**Comment:**

The paper tackles the imbalanced data learning issue, through the combination of two approaches:
* data augmentation based on non-linear independent representations
* causal modeling based on generalized contrastive learning.

The presented scheme is a quite complex mixture of recent ideas:
* generalized contrastive learning;
* non-linear independent sources (NICA)
* masked auto-regressive flow (MAF).

The approach proceeds mainly in 4 steps:
1. All non-minority classes (at least 2) are used to learn a feature encoder module $Z = e_\theta(X)$, and $Z$ supports a discriminant model for the non-minority classes;
2. NICA is used to build the sources $S = f_\psi(Z)$, using a masked auto-regressive flow, and ensuring that $f_\psi$ is invertible; its inverse $f_\psi^{-1}$ is viewed as the causal mechanism $S \rightarrow Z$;
3. An augmentation module is defined to generate new samples in the minority class by crossing-over the independent components of examples in the minority class (or using a parametric augmentation, e.g. using a Gaussian distribution fitted from the samples), subject to the assumption below;
4. A discriminant model based on $S$ is then trained for all classes.

The assumption done is that the causal mechanism ($Z = f_\psi^{-1}(S)$) is same for all classes.

The contribution on the top of generalized contrastive learning is to use an energy modeling based formulation and enforcing its smoothness using a MAF.

Another insightful ingredient consists in learning in the source space as opposed to, in the representation space.

The positioning and rationale of the approach, in supplementary material, is particularly interesting.
Besides, the authors did an excellent and thorough job in answering the reviewers concerns (regarding the limitations of the approach, the fact that the augmented examples are not guaranteed to fall in the minority class, the reproducibility of the results, the relationship with related work, e.g. Supervised Contrastive Learning (SCL) [Khosla, et al. (2020)], the fact that Z is frozen and not adapted after the augmentation).

While the paper contains more than one idea -- which is always dangerous -- it is the Area Chair's opinion that the material in it is sound, well thought from both theoretical and practical perspectives, and that it has the potential to inspire many NeurIPS attendees.